# Implicit Regularization for Optimal Sparse Recovery

**Tomas Vaškevičius[1], Varun Kanade[2], Patrick Rebeschini[1]**
[1] Department of Statistics, [2] Department of Computer Science
University of Oxford
{tomas.vaskevicius, patrick.rebeschini}@stats.ox.ac.uk
varunk@cs.ox.ac.uk

## Abstract

We investigate implicit regularization schemes for gradient descent methods applied to unpenalized least squares regression to solve the problem of reconstructing a sparse signal from an underdetermined system of linear measurements under the restricted isometry assumption. For a given parametrization yielding a non-convex optimization problem, we show that prescribed choices of initialization, step size and stopping time yield a statistically and computationally optimal algorithm that achieves the minimax rate with the same cost required to read the data up to poly-logarithmic factors. Beyond minimax optimality, we show that our algorithm adapts to instance difficulty and yields a dimension-independent rate when the signal-to-noise ratio is high enough. Key to the computational efficiency of our method is an increasing step size scheme that adapts to refined estimates of the true solution. We validate our findings with numerical experiments and compare our algorithm against explicit $\ell_1$ penalization. Going from hard instances to easy ones, our algorithm is seen to undergo a phase transition, eventually matching least squares with an oracle knowledge of the true support.

## 1 Introduction

Many problems in machine learning, science and engineering involve high-dimensional datasets where the dimensionality of the data $d$ is greater than the number of data points $n$. Linear regression with sparsity constraints is an archetypal problem in this setting. The goal is to estimate a $d$-dimensional vector $\mathbf{w}^\star \in \mathbb{R}^d$ with $k$ non-zero components from $n$ data points $(\mathbf{x}_i, y_i) \in \mathbb{R}^d \times \mathbb{R}$, $i \in \{1, \ldots, n\}$, linked by the linear relationship $y_i = \langle \mathbf{x}_i, \mathbf{w}^\star \rangle + \xi_i$, where $\xi_i$ is a possible perturbation to the $i^{th}$ observation. In matrix-vector form the model reads $\mathbf{y} = \mathbf{X}\mathbf{w}^\star + \xi$, where $\mathbf{x}_i$ corresponds to the $i^{th}$ row of the $n \times d$ design matrix $\mathbf{X}$. Over the past couple of decades, sparse linear regression has been extensively investigated from the point of view of both statistics and optimization.

In statistics, sparsity has been enforced by designing estimators with *explicit* regularization schemes based on the $\ell_1$ norm, such as the lasso [46] and the closely related basis pursuit [15] and Dantzig selector [13]. In the noiseless setting ($\xi = 0$), exact recovery is possible if and only if the design matrix satisfies the restricted nullspace property [16, 17, 19]. In the noisy setting ($\xi \neq 0$), exact recovery is not feasible and a natural criterion involves designing estimators $\widehat{\mathbf{w}}$ that can recover the minimax-optimal rate $k\sigma^2 \log(d/k)/n$ for the squared $\ell_2$ error $\|\widehat{\mathbf{w}} - \mathbf{w}^\star\|_2^2$ in the case of i.i.d. sub-Gaussian noise with variance proxy $\sigma^2$ when the design matrix satisfies restricted eigenvalue conditions [9, 48]. The lasso estimator, defined as any vector $\mathbf{w}$ that minimizes the objective $\|\mathbf{X}\mathbf{w} - \mathbf{y}\|_2^2 + \lambda\|\mathbf{w}\|_1$, achieves the minimax-optimal rate upon proper tuning of the regularization parameter $\lambda$. The restricted isometry property (RIP) [14] has been largely considered in the literature, as it implies both the restricted nullspace and eigenvalue conditions [16, 49], and as it is satisfied when the entries of $\mathbf{X}$ are i.i.d. sub-Gaussian and subexponential with sample size $n = \Omega(k \log(d/k))$ and $n = \Omega(k \log^2(d/k))$ respectively [32, 1], or when the columns are unitary, e.g. [23, 24, 39, 41].

In optimization, computationally efficient iterative algorithms have been designed to solve convex problems based on $\ell_1$ constraints and penalties, such as composite/proximal methods [4, 35]. Under restricted eigenvalue conditions, such as restricted strong convexity and restricted smoothness, various iterative methods have been shown to yield exponential convergence to the problem solution globally up to the statistical precision of the model [2], or locally once the iterates are close enough to the optimum and the support of the solution is identified [10, 28, 45]. In some regimes, for a prescribed choice of the regularization parameter, these algorithms are computationally efficient. They require $\widetilde{O}(1)$ iterations, where the notation $\widetilde{O}$ hides poly-logarithmic terms, and each iteration costs $O(nd)$. Hence the total running cost is $\widetilde{O}(nd)$, which is the cost to store/read the data in/from memory.

These results attest that there are regimes where optimal methods for sparse linear regression exist. However, these results reply upon tuning the hyperparameters for optimization, such as the step size, carefully, which in turn depends on identifying the correct hyperparameters, such as $\lambda$, for regularization. In practice, one has to resort to cross-validation techniques to tune the regularization parameter. Cross-validation adds an additional burden from a computational point of view, as the optimization algorithms need to be run for different choices of the regularization terms. In the context of linear regression with $\ell_2$ penalty, a.k.a. ridge regression, potential computational savings have motivated research on the design of *implicit* regularization schemes where model complexity is directly controlled by tuning the hyper-parameters of solvers applied to unpenalized/unconstrained programs, such as choice of initialization, step-size, iteration/training time. There has been increasing interest in understanding the effects of implicit regularization (sometimes referred to as implicit bias) of machine learning algorithms. It is widely acknowledged that the choice of algorithm, parametrization, and parameter-tuning, all affect the learning performance of models derived from training data. While implicit regularization has been extensively investigated in connection to the $\ell_2$ norm, there seem to be no results for sparse regression, which is surprising considering the importance of the problem.

## 1.1 Our Contributions

In this work, we merge statistics with optimization, and propose the first statistically and computationally optimal algorithm based on implicit regularization (initialization/step-size tuning and early stopping) for sparse linear regression under the RIP.

The algorithm that we propose is based on gradient descent applied to the unregularized, underdetermined objective function $\|\mathbf{X}\mathbf{w} - \mathbf{y}\|_2^2$ where $\mathbf{w}$ is parametrized as $\mathbf{w} = \mathbf{u} \odot \mathbf{u} - \mathbf{v} \odot \mathbf{v}$, with $\mathbf{u}, \mathbf{v} \in \mathbb{R}^d$ and $\odot$ denotes the coordinate-wise multiplication operator for vectors. This parametrization yields a non-convex problem in $\mathbf{u}$ and $\mathbf{v}$. We treat this optimization problem as a proxy to design a sequence of statistical estimators that correspond to the iterates of gradient descent applied to solve the sparse regression problem, and hence are cheap to compute iteratively. The matrix formulation of the same type of parametrization that we adopt has been recently considered in the setting of low-rank matrix recovery where it leads to exact recovery via implicit regularization in the *noiseless* setting under the RIP [25, 30]. In our case, this choice of parametrization yields an iterative algorithm that performs multiplicative updates on the coordinates of $\mathbf{u}$ and $\mathbf{v}$, in contrast to the additive updates obtained when gradient descent is run directly on the parameter $\mathbf{w}$, as in proximal methods. This feature allows us to reduce the convergence analysis to one-dimensional iterates and to differentiate the convergence on the support set $S = \{i \in \{1, \ldots, d\} : w_i^\star \neq 0\}$ from the convergence on its complement $S^c = \{1, \ldots, d\} \setminus S$.

We consider gradient descent initialized with $\mathbf{u}_0 = \mathbf{v}_0 = \alpha \mathbf{1}$, where $\mathbf{1}$ is the all-one vector. We show that with a sufficiently small initialization size $\alpha > 0$ and early stopping, our method achieves exact reconstruction with precision controlled by $\alpha$ in the noiseless setting, and minimax-optimal rates in the noisy setting. To the best of our knowledge, our results are the first to establish non-$\ell_2$ implicit regularization for a gradient descent method in a general *noisy* setting.[1] These results rely on a constant choice of step size $\eta$ that satisfies a bound related to the unknown parameter $w_{\max}^\star = \|\mathbf{w}^\star\|_\infty$. We show how this choice of $\eta$ can be derived from the data itself, i.e. only based on *known* quantities. If the noise vector $\xi$ is made up of i.i.d. sub-Gaussian components with variance proxy $\sigma^2$, this choice of $\eta$ yields $O((w_{\max}^\star \sqrt{n})/(\sigma \sqrt{\log d}) \log \alpha^{-1})$ iteration complexity to achieve minimax rates. In order to achieve *computational optimality*, we design a preconditioned version of gradient descent (on

the parameters **u** and **v**) that uses increasing step-sizes and has running time $\widetilde{O}(nd)$. The iteration-dependent preconditioner relates to the statistical nature of the problem. It is made by a sequence of diagonal matrices that implement a coordinate-wise increasing step-size scheme that allows different coordinates to accelerate convergence by taking larger steps based on refined estimates of the corresponding coordinates of $\mathbf{w}^\star$. This algorithm yields $O(\log((w_{\max}^\star \sqrt{n})/(\sigma \sqrt{\log d})) \log \alpha^{-1}$ iteration complexity to achieve minimax rates in the noisy setting. Since each iteration costs $O(nd)$, the total computation complexity is, up to poly-logarithmic factors, the same as simply storing/reading the data. This algorithm is minimax-optimal and, up to logarithmic factors, computationally optimal. In contrast, we are not aware of any work on implicit $\ell_2$ regularization that exploits an increasing step sizes scheme in order to attain computational optimality.

To support our theoretical results we present a simulation study of our methods and comparisons with the lasso estimator and with the gold standard oracle least squares estimator, which performs least squares regression on $S$ assuming oracle knowledge of it. We show that the number of iterations $t$ in our method plays a role similar to the lasso regularization parameter $\lambda$. Despite both algorithms being minimax-optimal with the right choice of $t$ and $\lambda$ respectively, the gradient descent optimization path—which is cheaper to compute as each iteration of gradient descent yields a new model—exhibits qualitative and quantitative differences from the lasso regularization path—which is more expensive to compute as each model requires solving a new lasso optimization program. In particular, the simulations emphasize how the multiplicative updates allow gradient descent to fit one coordinate of $\mathbf{w}^\star$ at a time, as opposed to the lasso estimator that tends to fit all coordinates at once. Beyond minimax results, we prove that our methods adapt to instance difficulty: for "easy" problems where the signal is greater than the noise, i.e. $w_{\min}^\star \gtrsim \|\mathbf{X}^\mathsf{T}\xi\|_\infty/n$ with $w_{\min}^\star = \min_{i \in S} |w_i^\star|$, our estimators achieve the statistical rate $k\sigma^2 \log(k)/n$, which does *not* depend on $d$. The experiments confirm this behavior and further attest that our estimators undergo a phase transition that is not observed for the lasso. Going from hard instances to easy ones, the learning capacity of implicitly-regularized gradient descent exhibits a qualitative transition and eventually matches the performance of oracle least squares.

## 1.2 Related Work

**Sparse Recovery.** The statistical properties of explicit $\ell_1$ penalization techniques are well studied [48, 13, 9, 31, 33]. Minimax rates for regression under sparsity constraints are derived in [37]. Computing the whole lasso regularization path can be done via the lars algorithm [18]. Another widely used approach is the glmnet which uses cyclic coordinate-descent with warm starts to compute regularization paths for generalized linear models with convex penalties on a pre-specified grid of regularization parameters [22]. [4] reviews various optimization techniques used in solving empirical risk minimization problems with sparsity inducing penalties. Using recent advances in mixed integer optimization, [8] shows that the best subset selection problem can be tackled for problems of moderate size. For such problem sizes, comparisons between the lasso and best subset selection problem ($\ell_0$ regularization) were recently made, suggesting that the best subset selection performs better in high signal-to-noise ratio regimes whereas the lasso performs better when the signal-to-noise ratio is low [29]. In this sense, our empirical study in Section 5 suggests that implicitly-regularized gradient descent is more similar to $\ell_0$ regularization than $\ell_1$ regularization. Several other techniques related to $\ell_1$ regularization and extensions to the lasso exist. We refer the interested reader to the books [11, 47].

**Implicit Regularization/Bias.** Connections between $\ell_2$ regularization and gradient descent optimization paths have been known for a long time and are well studied [12, 20, 52, 7, 38, 51, 34, 44, 3]. In contrast, the literature on implicit regularization inducing sparsity is scarce. Coordinate-descent optimization paths have been shown to be related to $\ell_1$ regularization paths in some regimes [21, 18, 40, 54]. Understanding such connections can potentially allow transferring the now well-understood theory developed for penalized forms of regularization to early-stopping-based regularization which can result in lower computational complexity. Recently, [53] have shown that neural networks generalize well even without explicit regularization despite the capacity to fit unstructured noise. This suggests that some implicit regularization effect is limiting the capacity of the obtained models along the optimization path and thus explaining generalization on structured data. Understanding such effects has recently drawn a lot of attention in the machine learning community. In particular, it is now well understood that the optimization algorithm itself can be biased towards a particular set of solutions for underdetermined problems with many global minima where, in contrast

to the work cited above, the bias of optimization algorithm is investigated at or near convergence, usually in a noiseless setting [43, 27, 26, 25, 30]. We compare our assumptions with the ones made in [30] in Appendix G.

**Remark 1** (Concurrent Work). *After completing this work we became aware of independent concurrent work [56] which considers Hadamard product reparametrization $\mathbf{w}_t = \mathbf{u}_t \odot \mathbf{v}_t$ in order to implicitly induce sparsity for linear regression under the RIP assumption. Our work is significantly different in many aspects discussed in Appendix H. In particular, we obtain computational optimality and can properly handle the general noisy setting.*

## 2 Model and Algorithms

We consider the model defined in the introduction. We denote vectors with boldface letters and real numbers with normal font; thus, $\mathbf{w}$ denotes a vector and $w_i$ denotes the $i^{th}$ coordinate of $\mathbf{w}$. For any index set $A$ we let $\mathbf{1}_A$ denote a vector that has a 1 entry in all coordinates $i \in A$ and a 0 entry elsewhere. We denote coordinate-wise inequalities by $\preccurlyeq$. With a slight abuse of notation we write $\mathbf{w}^2$ to mean the vector obtained by squaring each component of $\mathbf{w}$. Finally, we denote inequalities up to multiplicative absolute constants, meaning that they do not depend on any parameters of the problem, by $\lesssim$. A table of notation can be found in Appendix J.

We now define the restricted isometry property which is the key assumption in our main theorems.

**Definition 1** (Restricted Isometry Property (RIP)). *A $n \times d$ matrix $\mathbf{X}/\sqrt{n}$ satisfies the $(\delta, k)$-(RIP) if for any $k$-sparse vector $\mathbf{w} \in \mathbb{R}^d$ we have $(1 - \delta) \|\mathbf{w}\|_2^2 \leq \|\mathbf{X}\mathbf{w}/\sqrt{n}\|_2^2 \leq (1 + \delta) \|\mathbf{w}\|_2^2$.*

The RIP assumption was introduced in [14] and is standard in the compressed sensing literature. It requires that all $n \times k$ sub-matrices of $\mathbf{X}/\sqrt{n}$ are approximately orthonormal where $\delta$ controls extent to which this approximation holds. Checking if a given matrix satisfies the RIP is NP-hard [5]. In compressed sensing applications the matrix $\mathbf{X}/\sqrt{n}$ corresponds to how we measure signals and it can be chosen by the designer of a sparse-measurement device. Random matrices are known to satisfy the RIP with high probability, with $\delta$ decreasing to 0 as $n$ increases for a fixed $k$ [6].

We consider the following problem setting. Let $\mathbf{u}, \mathbf{v} \in \mathbb{R}^d$ and define the mean squared loss as

$$\mathcal{L}(\mathbf{u}, \mathbf{v}) = \frac{1}{n} \|\mathbf{X} (\mathbf{u} \odot \mathbf{u} - \mathbf{v} \odot \mathbf{v}) - \mathbf{y}\|_2^2.$$

Letting $\mathbf{w} = \mathbf{u} \odot \mathbf{u} - \mathbf{v} \odot \mathbf{v}$ and performing gradient descent updates on $\mathbf{w}$, we recover the original parametrization of mean squared error loss which does not implicitly induce sparsity. Instead, we perform gradient descent updates on $(\mathbf{u}, \mathbf{v})$ treating it as a vector in $\mathbb{R}^{2d}$ and we show that the corresponding optimization path contains sparse solutions.

Let $\eta > 0$ be the learning rate, $(\mathbf{m}_t)_{t \geq 0}$ be a sequence of vectors in $\mathbb{R}^d$ and $\mathrm{diag}(\mathbf{m}_t)$ be a $d \times d$ diagonal matrix with $\mathbf{m}_t$ on its diagonal. We consider the following general form of gradient descent:

$$(\mathbf{u}_{t+1}, \mathbf{v}_{t+1}) = (\mathbf{u}_t, \mathbf{v}_t) - \eta \, \mathrm{diag}(\mathbf{m}_t, \mathbf{m}_t) \frac{\partial \mathcal{L}(\mathbf{u}_t, \mathbf{v}_t)}{\partial(\mathbf{u}_t, \mathbf{v}_t)}. \tag{1}$$

We analyze two different choices of sequences $(\mathbf{m}_t)_{t \geq 0}$ yielding two separate algorithms.

**Algorithm 1.** *Let $\alpha, \eta > 0$ be two given parameters. Let $\mathbf{u}_0 = \mathbf{v}_0 = \alpha$ and for all $t \geq 0$ we let $\mathbf{m}_t = \mathbf{1}$. Perform the updates given in (1).*

**Algorithm 2.** *Let $\alpha, \tau \in \mathbb{N}$ and $w_{\max}^\star \leq \hat{z} \leq 2w_{\max}^\star$ be three given parameters. Set $\eta = \frac{1}{20\hat{z}}$ and $\mathbf{u}_0 = \mathbf{v}_0 = \alpha$. Perform the updates in (1) with $\mathbf{m}_0 = \mathbf{1}$ and $\mathbf{m}_t$ adaptively defined as follows:*

1. *Set $\mathbf{m}_t = \mathbf{m}_{t-1}$.*

2. *If $t = m\tau \lceil \log \alpha^{-1} \rceil$ for some natural number $m \geq 2$ then let $m_{t,j} = 2m_{t-1,j}$ for all $j$ such that $u_{t,j}^2 \vee v_{t,j}^2 \leq 2^{-m-1}\hat{z}$.*

Algorithm 1 corresponds to gradient descent with a constant step size, whereas Algorithm 2 doubles the step-sizes for small enough coordinates after every $\tau \lceil \log \alpha^{-1} \rceil$ iterations.

Before stating the main results we define some key quantities. First, our results are sensitive to the condition number $\kappa = \kappa(\mathbf{w}^\star) = w_{\max}^\star / w_{\min}^\star$ of the true parameter vector $\mathbf{w}^\star$. Since we are not able

to recover coordinates below the maximum noise term $\|\mathbf{X}^\mathsf{T}\xi\|_\infty/n$, for a desired precision $\varepsilon$ we can treat all coordinates of $\mathbf{w}^\star$ below $\varepsilon \vee (\|\mathbf{X}^\mathsf{T}\xi\|_\infty/n)$ as 0. This motivates the following definition of an effective condition number for given $\mathbf{w}^\star, \mathbf{X}, \xi$ and $\varepsilon$:

$$\kappa^{\mathrm{eff}} = \kappa^{\mathrm{eff}}(\mathbf{w}^\star, \mathbf{X}, \xi, \varepsilon) = w_{\max}^\star/(w_{\min}^\star \vee \varepsilon \vee (\|\mathbf{X}^\mathsf{T}\xi\|_\infty/n)).$$

We remark that $\kappa^{\mathrm{eff}}(\mathbf{w}^\star, \mathbf{X}, \xi, \varepsilon) \leq \kappa(\mathbf{w}^\star)$. Second, we need to put restrictions on the RIP constant $\delta$ and initialization size $\alpha$. These restrictions are given by the following:

$$\delta(k, \mathbf{w}^\star, \mathbf{X}, \xi, \varepsilon) = 1/(\sqrt{k}(1 \vee \log \kappa^{\mathrm{eff}}(\mathbf{w}^\star))), \quad \alpha(\mathbf{w}^\star, \varepsilon, d) \coloneqq \frac{\varepsilon^2 \wedge \varepsilon \wedge 1}{(2d+1)^2 \vee (w_{\max}^\star)^2} \wedge \frac{\sqrt{w_{\min}^\star}}{2}.$$

## 3   Main Results

The following result is the backbone of our contributions. It establishes rates for Algorithm 1 in the $\ell_\infty$ norm as opposed to the typical rates for the lasso that are often only derived for the $\ell_2$ norm.

**Theorem 1.** *Fix any $\varepsilon > 0$. Suppose that $\mathbf{X}/\sqrt{n}$ satisfies the $(k+1, \delta)$-RIP with $\delta \lesssim \delta(k, \mathbf{w}^\star, \mathbf{X}, \xi, \varepsilon)$ and let the initialization $\alpha$ satisfy $\alpha \leq \alpha(\mathbf{w}^\star, \varepsilon, d)$. Then, Algorithm 1 with $\eta \leq 1/(20w_{\max}^\star)$ and $t = O((\kappa^{\mathrm{eff}}(\mathbf{w}^\star))/(\eta w_{\max}^\star) \log \alpha^{-1})$ iterations satisfies*

$$|w_{t,i} - w_i^\star| \lesssim \begin{cases} \left\|\frac{1}{n}\mathbf{X}^\mathsf{T}\xi\right\|_\infty \vee \varepsilon & \text{if } i \in S \text{ and } w_{\min}^\star \lesssim \left\|\frac{1}{n}\mathbf{X}^\mathsf{T}\xi\right\|_\infty \vee \varepsilon, \\ \left|\frac{1}{n}\left(\mathbf{X}^\mathsf{T}\xi\right)_i\right| \vee \delta\sqrt{k}\left\|\frac{1}{n}\mathbf{X}^\mathsf{T}\xi \odot \mathbf{1}_S\right\|_\infty \vee \varepsilon & \text{if } i \in S \text{ and } w_{\min}^\star \gtrsim \left\|\frac{1}{n}\mathbf{X}^\mathsf{T}\xi\right\|_\infty \vee \varepsilon, \\ \sqrt{\alpha} & \text{if } i \notin S. \end{cases}$$

This result shows how the parameters $\alpha, \eta$ and $t$ affect the learning performance of gradient descent. The size of $\alpha$ controls the size of the coordinates outside the true support $S$ at the stopping time. We discuss the role and also the necessity of small initialization size to achieve the desired statistical performance in Section 5. A different role is played by the step size $\eta$ whose size affects the optimal stopping time $t$. In particular, $(\eta t)/\log \alpha^{-1}$ can be seen as a regularization parameter closely related to $\lambda^{-1}$ for the lasso. To see this, suppose that the noise $\xi$ is $\sigma^2$-sub-Gaussian with independent components. Then with high probability $\|\mathbf{X}^\mathsf{T}\xi\|_\infty/n \lesssim (\sigma\sqrt{\log d})/\sqrt{n})$. In such a setting an optimal choice of $\lambda$ for the lasso is $\Theta((\sigma\sqrt{\log d})/\sqrt{n})$. On the other hand, letting $t^\star$ be the optimal stopping time given in Theorem 1, we have $(\eta t^\star)/\log \alpha^{-1} = O(1/w_{\min}^\star(\mathbf{X}, \xi, \varepsilon)) = O(\sqrt{n}/(\sigma\sqrt{\log d}))$.

The condition $\eta \leq 1/(20w_{\max}^\star)$ is also necessary up to constant factors in order to prevent explosion. If we can set $1/w_{\max}^\star \lesssim \eta \leq 1/(20w_{\max}^\star)$ then the iteration complexity of Theorem 1 reduces to $O(\kappa^{\mathrm{eff}}(\mathbf{w}^\star)\log \alpha^{-1})$. The magnitude of $w_{\max}^\star$ is, however, an unknown quantity. Similarly, setting the proper initialization size $\alpha$ depends on $w_{\max}^\star, w_{\min}^\star, d$ and the desired precision $\varepsilon$. The requirement that $\alpha \leq \sqrt{w_{\min}^\star}/2$ is an artifact of our proof technique and tighter analysis could replace this condition by simply $\alpha \leq \varepsilon$. Hence the only unknown quantity for selecting a proper initialization size is $w_{\max}^\star$.

The next theorem shows how $w_{\max}^\star$ can be estimated from the data up to a multiplicative factor 2 at the cost of one gradient descent iteration. Once this estimate is computed, we can properly set the initialization size and the learning rate $\eta \asymp \frac{1}{w_{\max}^\star}$ which satisfies our theory and is tight up to constant multiplicative factors. We remark that $\tilde{\eta}$ used in Theorem 2 can be set arbitrarily small (e.g., set $\tilde{\eta} = 10^{-10}$) and is only used for one gradient descent step in order to estimate $w_{\max}^\star$.

**Theorem 2** (Estimating $w_{\max}^\star$). *Set $\alpha = 1$ and suppose that $\mathbf{X}/\sqrt{n}$ satisfies the $(k+1, \delta)$-RIP with $\delta \leq 1/(20\sqrt{k})$. Let the step size $\tilde{\eta}$ be any number satisfying $0 < \tilde{\eta} \leq 1/(5w_{\max}^\star)$ and suppose that $w_{\max}^\star \geq 5\|\mathbf{X}^\mathsf{T}\xi\|_\infty/n$. Perform one step of gradient descent and for each $i \in \{1, \ldots, d\}$ compute the update factors defined as $f_i^+ = (u_1)_i$ and $f_i^- = (v_1)_i$. Let $f_{\max} = \|\mathbf{f}^+\|_\infty \vee \|\mathbf{f}^-\|_\infty$. Then $w_{\max}^\star \leq (f_{\max} - 1)/(3\tilde{\eta}) < 2w_{\max}^\star$.*

We present three main corollaries of Theorem 1. The first one shows that in the noiseless setting exact recovery is possible and is controlled by the desired precision $\varepsilon$ and hence by the initialization size $\alpha$.

**Corollary 1** (Noiseless Recovery). *Let $\xi = 0$. Under the assumptions of Theorem 1, the choice of $\eta$ given by Theorem 2 and $t = O(\kappa^{\mathrm{eff}}(\mathbf{w}^\star) \log \alpha^{-1})$, Algorithm 1 yields $\|\mathbf{w}_t - \mathbf{w}^\star\|_2^2 \lesssim k\varepsilon^2$.*

In the general noisy setting exact reconstruction of $\mathbf{w}^\star$ is not possible. In fact, the bounds in Theorem 1 do not improve with $\varepsilon$ chosen below the maximum noise term $\|\mathbf{X}^\mathsf{T}\xi\|_\infty/n$. In the following corollary we show that with a small enough $\varepsilon$ if the design matrix $\mathbf{X}$ is fixed and the noise vector $\xi$ is sub-Gaussian, we recover minimax-optimal rates for $\ell_2$ error. Our error bound is minimax-optimal in the setting of sub-linear sparsity, meaning that there exists a constant $\gamma > 1$ such that $k^\gamma \leq d$.

**Corollary 2** (Minimax Rates in the Noisy Setting). *Let the noise vector $\xi$ be made of independent $\sigma^2$-sub-Gaussian entries. Let $\varepsilon = 4\sqrt{\sigma^2 \log(2d)}/\sqrt{n}$. Under the assumptions of Theorem 1, the choice of $\eta$ given by Theorem 2 and $t = O(\kappa^{\text{eff}}(\mathbf{w}^\star) \log \alpha^{-1}) = O((w^\star_{\max}\sqrt{n})/(\sigma\sqrt{\log d})\log \alpha^{-1})$, Algorithm 1 yields $\|\mathbf{w}_t - \mathbf{w}^\star\|_2^2 \lesssim (k\sigma^2 \log d)/n$ with probability at least $1 - 1/(8d^3)$.*

The next corollary states that gradient descent automatically adapts to the difficulty of the problem. The statement of Theorem 1 suggests that our bounds undergo a phase-transition when $w^\star_{\min} \gtrsim \|\mathbf{X}^\mathsf{T}\xi\|_\infty/n$ which is also supported by our empirical findings in Section 5. In the $\sigma^2$-sub-Gaussian noise setting the transition occurs as soon as $n \gtrsim (\sigma^2 \log d)/(w^\star_{\min})^2$. As a result, the statistical bounds achieved by our algorithm are independent of $d$ in such a setting. To see that, note that while the term $\|\mathbf{X}^\mathsf{T}\xi\|_\infty/n$ grows as $O(\log d)$, the term $\|\mathbf{X}^\mathsf{T}\xi \odot \mathbf{1}_S\|_\infty/n$ grows only as $O(\log k)$. In contrast, performance of the lasso deteriorates with $d$ regardless of the difficulty of the problem. We illustrate this graphically and give a theoretical explanation in Section 5. We remark that the following result does not contradict minimax optimality because we now treat the true parameter $\mathbf{w}^\star$ as fixed.

**Corollary 3** (Instance Adaptivity). *Let the noise vector $\xi$ be made of independent $\sigma^2$-sub-Gaussian entries. Let $\varepsilon = 4\sqrt{\sigma^2 \log(2k)}/\sqrt{n}$. Under the assumptions of Theorem 1, the choice of $\eta$ given by Theorem 2 and $t = O(\kappa^{\text{eff}}(\mathbf{w}^\star)\log \alpha^{-1}) = O((w^\star_{\max}\sqrt{n})/(\sigma\sqrt{\log k})\log \alpha^{-1})$, Algorithm 1 yields $\|\mathbf{w}_t - \mathbf{w}^\star\|_2^2 \lesssim (k\sigma^2 \log k)/n$. with probability at least $1 - 1/(8k^3)$.*

The final theorem we present shows that the same statistical bounds achieved by Algorithm 1 are also attained by Algorithm 2. This algorithm is not only optimal in a statistical sense, but it is also optimal computationally up to poly-logarithmic factors.

**Theorem 3.** *Compute $\hat{z}$ using Theorem 2. Under the setting of Theorem 1 there exists a large enough absolute constant $\tau$ so that Algorithm 2 parameterized with $\alpha, \tau$ and $\hat{z}$ satisfies the result of Theorem 1 and $t = O(\log \kappa^{\text{eff}} \log \alpha^{-1})$ iterations.*

Corollaries 1, 2 and 3 also hold for Algorithm 2 with stopping time equal to $O(\log \kappa^{\text{eff}} \log \alpha^{-1})$. We emphasize that both Theorem 1 and 3 use gradient-based updates to obtain a sequence of models with optimal statistical properties instead of optimizing the objective function $\mathcal{L}$. In fact, if we let $t \to \infty$ for Algorithm 2 the iterates would explode.

## 4 Proof Sketch

In this section we prove a simplified version of Theorem 1 under the assumption $\mathbf{X}^\mathsf{T}\mathbf{X}/n = \mathbf{I}$. We further highlight the intricacies involved in the general setting and present the intuition behind the key ideas there. The gradient descent updates on $\mathbf{u}_t$ and $\mathbf{v}_t$ as given in (1) can be written as

$$\mathbf{u}_{t+1} = \mathbf{u}_t \odot \left(\mathbf{1} - (4\eta/n)\mathbf{X}^\mathsf{T}(\mathbf{X}\mathbf{w}_t - \mathbf{y})\right), \quad \mathbf{v}_{t+1} = \mathbf{v}_t \odot \left(\mathbf{1} + (4\eta/n)\mathbf{X}^\mathsf{T}(\mathbf{X}\mathbf{w}_t - \mathbf{y})\right). \quad (2)$$

The updates can be succinctly represented as $\mathbf{u}_{t+1} = \mathbf{u}_t \odot (\mathbf{1} - \mathbf{r})$ and $\mathbf{v}_{t+1} = \mathbf{v}_t \odot (\mathbf{1} + \mathbf{r})$, where by our choice of $\eta$, $\|\mathbf{r}\|_\infty \leq 1$. Thus, $(\mathbf{1} - \mathbf{r}) \odot (\mathbf{1} + \mathbf{r}) \preccurlyeq \mathbf{1}$ and we have $\mathbf{u}_t \odot \mathbf{v}_t \preccurlyeq \mathbf{u}_0 \odot \mathbf{v}_0 = \alpha^2 \mathbf{1}$. Hence for any $i$, only one of $|u_{t,i}|$ and $|v_{t,i}|$ can be larger then the initialization size while the other is effectively equal to 0. Intuitively, $u_{t,i}$ is used if $w_i^\star > 0$, $v_{t,i}$ if $w_i^\star < 0$ and hence one of these terms can be merged into an error term $b_{t,i}$ as defined below. The details appear in Appendix B.4. To avoid getting lost in cumbersome notation, in this section we will assume that $\mathbf{w}^\star \succcurlyeq 0$ and $\mathbf{w} = \mathbf{u} \odot \mathbf{u}$.

**Theorem 4.** *Assume that $\mathbf{w}^\star \succcurlyeq 0$, $\frac{1}{n}\mathbf{X}^\mathsf{T}\mathbf{X} = \mathbf{I}$, and that there is no noise ($\xi = 0$). Parameterize $\mathbf{w} = \mathbf{u} \odot \mathbf{u}$ with $\mathbf{u}_0 = \alpha\mathbf{1}$ for some $0 < \alpha < \sqrt{w^\star_{\min}}$. Letting $\eta \leq 1/(10w^\star_{\max})$ and $t = O(\log(w^\star_{\max}/\alpha^2)/(\eta w^\star_{\min}))$, Algorithm 1 yields $\|\mathbf{w}_t - \mathbf{w}^\star\|_\infty \leq \alpha^2$.*

*Proof.* As $\mathbf{X}^\mathsf{T}\mathbf{X}/n = \mathbf{I}, \mathbf{y} = \mathbf{X}\mathbf{w}^\star$, and $\mathbf{v}_t = 0$, the updates given in equation (2) reduce component-wise to updates on $\mathbf{w}_t$ given by $w_{t+1,i} = w_{t,i} \cdot (1 - 4\eta(w_{t,i} - w_i^\star))^2$. For $i$ such that $w_i^\star = 0$, $w_{t,i}$

is non-increasing and hence stays below $\alpha^2$. For $i$ such that $w_i^\star > \alpha^2$, the update rule given above ensures that as long as $w_{t,i} < w_i^\star/2$, $w_{t,i}$ increases at an exponential rate with base at least $(1+2\eta w_i^\star)^2$. As $w_{0,i} = \alpha^2$, in $O(\log(w_i^\star/\alpha^2)/(\eta w_i^\star))$ steps, it holds that $w_{t,i} \geq w_i^\star/2$. Subsequently, the gap $(w_i^\star - w_{t,i})$ halves every $O(1/\eta w_i^\star)$ steps; thus, in $O(\log(w_{\max}^\star/\alpha^2)/(\eta w_{\min}^\star))$ steps we have $\|\mathbf{w}_t - \mathbf{w}^\star\|_\infty \leq \alpha^2$. The exact details are an exercise in calculus, albeit a rather tedious one, and appear in Appendix B.1. □

The proof of Theorem 4 contains the key ideas of the proof of Theorem 1. However, the presence of noise ($\xi \neq 0$) and only having *restricted isometry* of $\mathbf{X}^\mathsf{T}\mathbf{X}$ rather than *isometry* requires a subtle and involved analysis. We remark that we can prove tighter bounds in Theorem 4 than the ones in Theorem 1 because we are working in a simplified setting.

**Error Decompositions.** We decompose $\mathbf{w}_t$ into $\mathbf{s}_t := \mathbf{w}_t \odot \mathbf{1}_S$ and $\mathbf{e}_t := \mathbf{w}_t \odot \mathbf{1}_{S^c}$ so that $\mathbf{w}_t = \mathbf{s}_t + \mathbf{e}_t$. We define the following error sequences:

$$\mathbf{b}_t = \mathbf{X}^\mathsf{T}\mathbf{X}\mathbf{e}_t/n + \mathbf{X}^\mathsf{T}\xi/n, \qquad \mathbf{p}_t = \left(\mathbf{X}^\mathsf{T}\mathbf{X}/n - \mathbf{I}\right)(\mathbf{s}_t - \mathbf{w}^\star),$$

which allows us to write updates on $\mathbf{s}_t$ and $\mathbf{e}_t$ as

$$\mathbf{s}_{t+1} = \mathbf{s}_t \odot (1 - 4\eta(\mathbf{s}_t - \mathbf{w}^* + \mathbf{p}_t + \mathbf{b}_t))^2, \qquad \mathbf{e}_{t+1} = \mathbf{e}_t \odot (1 - 4\eta(\mathbf{p}_t + \mathbf{b}_t))^2.$$

**Error Sequence $\mathbf{b}_t$.** Since our theorems require stopping before $\|\mathbf{e}_t\|_\infty$ exceeds $\sqrt{\alpha}$, the term $\mathbf{X}^\mathsf{T}\mathbf{X}\mathbf{e}_t/n$ can be controlled entirely by the initialization size. Hence $\mathbf{b}_t \approx \mathbf{X}^\mathsf{T}\xi/n$ and it represents an *irreducible* error arising due to the noise on the labels. For any $i \in S$ at stopping time $t$ we cannot expect the error on the $i^{th}$ coordinate $|w_{t,i} - w_i^\star|$ to be smaller than $|(\mathbf{X}^\mathsf{T}\xi)/n)_i|$. If we assume $\mathbf{p}_t = 0$ and $\xi \neq 0$ then in light of our simplified Theorem 4 we see that the terms in $\mathbf{e}_t$ grow exponentially with base at most $(1 + 4\eta\|\mathbf{X}^\mathsf{T}\xi\|_\infty/n)$. We can fit all the terms in $\mathbf{s}_t$ such that $|w_i^\star| \gtrsim \|\mathbf{X}^\mathsf{T}\xi\|_\infty/n$ which leads to minimax-optimal rates. Moreover, if $w_{\min}^\star \gtrsim \|\mathbf{X}^\mathsf{T}\xi\|_\infty/n$ then all the elements in $\mathbf{s}_t$ grow exponentially at a faster rate than all of the error terms. This corresponds to the easy setting where the resulting error depends only on $\|\mathbf{X}^\mathsf{T}\xi/n \odot \mathbf{1}_S\|_\infty$ yielding dimension-independent error bounds. For more details see Appendix B.2.

**Error Sequence $\mathbf{p}_t$.** Since $\mathbf{s}_t - \mathbf{w}^\star$ is a $k$-sparse vector using the RIP we can upper-bound $\|\mathbf{p}_t\|_\infty \leq \sqrt{k}\delta \|\mathbf{s}_t - \mathbf{w}^\star\|_\infty$. Note that for small $t$ we have $\|\mathbf{s}_0\|_\infty \approx \alpha^2 \approx 0$ and hence, ignoring the logarithmic factor in the definition of $\delta$ in the worst case we have $Cw_{\max}^\star \leq \|\mathbf{p}_t\|_\infty < w_{\max}^\star$ for some absolute constant $0 < C < 1$. If $w_{\max}^\star \gg \|\mathbf{X}^\mathsf{T}\xi\|_\infty/n$ then the error terms can grow exponentially with base $(1 + 4\eta \cdot Cw_{\max}^\star)$ whereas the signal terms such that $|w_i^\star| \ll w_{\max}^\star$ can shrink exponentially at rate $(1 - 4\eta \cdot Cw_{\max}^\star)$. On the other hand, in the light of Theorem 4 the signal elements converge exponentially fast to the true parameters $w_{\max}^\star$ and hence the error sequence $\mathbf{p}_t$ should be exponentially decreasing. For small enough $C$ and a careful choice of initialization size $\alpha$ we can ensure that elements of $\mathbf{p}_t$ decrease before the error components in $\mathbf{e}_t$ get too large or the signal components in $\mathbf{s}_t$ get too small. For more details see Appendix A.2 and B.3.

**Tuning Learning Rates.** The proof of Theorem 2 is given in Appendix D. If we choose $1/w_{\max}^\star \lesssim \eta \leq 1/(10w_{\max}^\star)$ in Theorem 4 then all coordinates converge in $O(\kappa \log(w_{\max}^\star/\alpha^2))$ iterations. The reason the factor $\kappa$ appears is the need to ensure that the convergence of the component $w_i^\star = w_{\max}^\star$ is stable. However, this conservative setting of the learning rate unnecessarily slows down the convergence for components with $w_i^\star \ll w_{\max}^\star$. In Theorem 4, oracle knowledge of $\mathbf{w}^\star$ would allow to set an individual step size for each coordinate $i \in S$ equal to $\eta_i = 1/(10w_i^\star)$ yielding the total number of iterations equal to $O(\log(w_{\max}^\star/\alpha^2))$. In the setting where $\mathbf{X}^\mathsf{T}\mathbf{X}/n \neq \mathbf{I}$ this would not be possible even with the knowledge of $\mathbf{w}^\star$, since the error sequence $\mathbf{p}_t$ can be initially too large which would result in explosion of the coordinates $i$ with $|w_i^\star| \ll w_{\max}^\star$. Instead, we need to wait for $\mathbf{p}_t$ to get small enough before we increase the step size for some of the coordinates as described in Algorithm 2. The analysis is considerably involved and the full proof can be found in Appendix E. We illustrate effects of increasing step sizes in Section 5.

## 5  Simulations

Unless otherwise specified, the default simulation set up is as follows. We let $\mathbf{w}^\star = \gamma\mathbf{1}_S$ for some constant $\gamma$. For each run the entries of $\mathbf{X}$ are sampled as i.i.d. Rademacher random variables and the noise vector $\xi$ follows i.i.d. $N(0, \sigma^2)$ distribution. For $\ell_2$ plots each simulation is repeated a

total of 30 times and the median $\ell_2$ error is depicted. The error bars in all the plots denote the $25^{th}$ and $75^{th}$ percentiles. Unless otherwise specified, the default values for simulation parameters are $n = 500, d = 10^4, k = 25, \alpha = 10^{-12}, \gamma = 1, \sigma = 1$ and for Algorithm 2 we set $\tau = 10$.

**Effects of Initialization Size.** As discussed in Section 4 each coordinate grows exponentially at a different rate. In Figure 1 we illustrate the necessity of small initialization for bringing out the exponential nature of coordinate paths allowing to effectively fit them one at a time. For more intuition, suppose that coordinates outside the true support grow at most as fast as $(1 + \varepsilon)^t$ while the coordinates on the true support grow at least as fast as $(1 + 2\varepsilon)^t$. Since exponential function is very sensitive to its base, for large enough $t$ we have $(1 + \varepsilon)^t \ll (1 + 2\varepsilon)^t$. The role of the initialization size $\alpha$ is then finding a small enough $\alpha$ such that for large enough $t$ we have $\alpha^2(1 + \varepsilon)^t \approx 0$ while $\alpha^2(1 + 2\varepsilon)^t$ is large enough to ensure convergence of the coordinates on the true support.

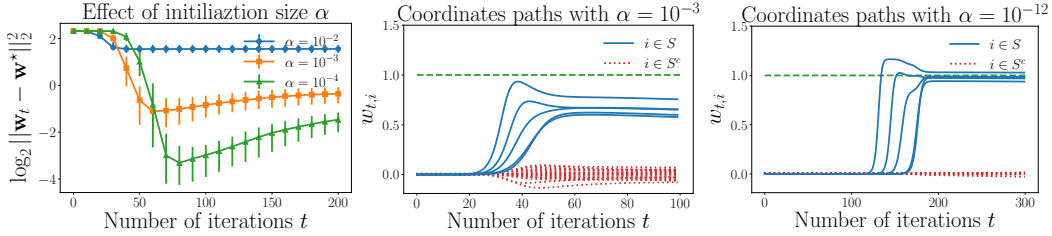

Figure 1: Effects of initialization size. We set $k = 5, n = 100, \eta = 0.05, \sigma = 0.5$ and run Algorithm 1. We remark that the $X$ axes in the two figures on the right differ due to different choices of $\alpha$.

**Exponential Convergence with Increasing Step Sizes.** We illustrate the effects of Algorithm 2 on an ill-conditioned target with $\kappa = 64$. Algorithm 1 spends approximately twice the time to fit each coordinate that the previous one, which is expected, since the coordinate sizes decrease by half. On the other hand, as soon as we increase the corresponding step size, Algorithm 2 fits each coordinate at approximately the same number of iterations, resulting in $O(\log \kappa \log \alpha^{-1})$ total iterations. Figure 2 confirms this behavior in simulations.

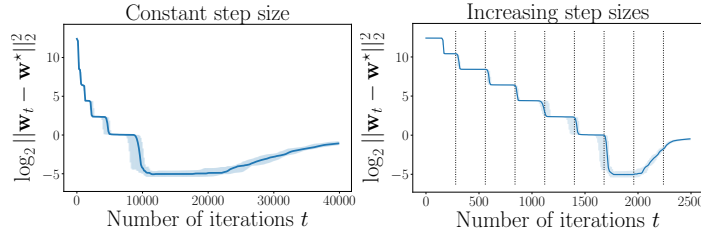

Figure 2: Comparison of Algorithms 1 and 2. Let $n = 250, k = 7$ and the non-zero coordinates of $\mathbf{w}^\star$ be $\{2^i : i = 0, \dots, 6\}$. For both algorithms let $\eta = 1/(20 \cdot 2^6)$. The vertical lines in the figure on the right are equally spaced at $\tau \log \alpha^{-1}$ iterations. The scale of x axes differ by a factor of 16. The shaded region corresponds to $25^{th}$ and $75^{th}$ percentiles over 30 runs.

**Phase Transitions.** As suggested by our main results, we present empirical evidence that when $w_{\min}^\star \gtrsim \|\mathbf{X}^\mathsf{T}\xi\|_\infty/n$ our algorithms undergo a phase transition with dimension-independent error bounds. We plot results for three different estimators. First we run Algorithm 2 for 2000 iterations and save every $10^{th}$ model. Among the 200 obtained models we choose the one with the smallest error on a *validation* dataset of size $n/4$. We run the lasso for 200 choices of $\lambda$ equally spaced on a logarithmic scale and for each run we select a model with the smallest $\ell_2$ parameter estimation error using an *oracle knowledge* of $\mathbf{w}^\star$. Finally, we perform a least squares fit using an *oracle knowledge* of the true support $S$. Figure 3 illustrates, that with varying $\gamma, \sigma$ and $n$ we can satisfy the condition $w_{\min}^\star \gtrsim \|\mathbf{X}^\mathsf{T}\xi\|_\infty/n$ at which point our method approaches an oracle-like performance. Given exponential nature of the coordinate-wise convergence, all coordinates of the true support grow at a strictly larger exponential rate than all of the coordinates on $S^c$ as soon as $w_{\min}^\star - \|\mathbf{X}^\mathsf{T}\xi\|_\infty/n > \|\mathbf{X}^\mathsf{T}\xi\|_\infty/n$. An approximate solution of this equation is shown in Figure 3 using vertical red lines.

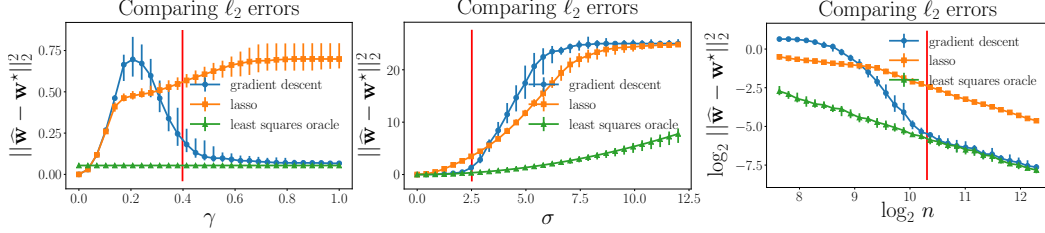

Figure 3: Phase transitions. The figure on the right uses $\gamma = 1/4$. The red vertical lines show solutions of the equation $w^\star_{\min} = \gamma = 2 \cdot \mathbb{E}\left[ \|\mathbf{X}^{\mathsf{T}}\xi\|_\infty / n \right] \leq 2 \cdot \sigma \sqrt{2\log(2d)}/\sqrt{n}$.

**Dimension Free Bounds in the Easy Setting.** Figure 4 shows that when $w^\star_{\min} \gtrsim \|\mathbf{X}^{\mathsf{T}}\xi\|_\infty / n$ our algorithm matches the performance of oracle least squares which is independent of $d$. In contrast, the performance of the lasso deteriorates as $d$ increases. To see why this is the case, in the setting where $\mathbf{X}^{\mathsf{T}}\mathbf{X}/n = \mathbf{I}$, the lasso solution with parameter $\lambda$ has a closed form solution $w_i^\lambda = \mathrm{sign}(w_i^{\mathsf{LS}})(|w_i^{\mathsf{LS}}| - \lambda)_+$, where $\mathbf{w}^{\mathsf{LS}}$ is the least squares solution. In the sub-Gaussian noise setting, the minimax rates are achieved by the choice $\lambda = \Theta(\sqrt{\sigma^2 \log(d)/n})$ introducing a bias which depends on $\log d$. Such a bias is illustrated in Figure 4 and is not present at the optimal stopping time of our algorithm.

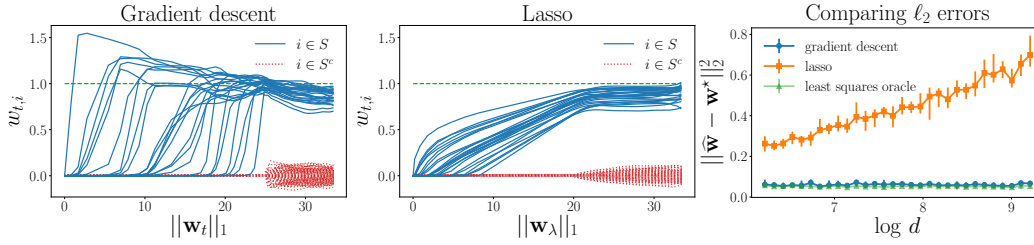

Figure 4: Dimension dependent bias for the lasso. In contrast, in a high signal-to-noise ratio setting gradient descent is able to recover coordinates on $S$ without a visible bias.

# 6   Summary and Further Improvements

In this paper, we have provided the first statistical and computational guarantees for two algorithms based on implicit regularization applied to a sparse recovery problem under the RIP assumption in the general noisy setting. We show that Algorithms 1 and 2 yield optimal statistical rates and, in contrast to $\ell_1$-penalization, adapt to the problem difficulty. In particular, given enough data both algorithms yield dimension-independent rates. While our algorithms are parametrized by step-size, initialization-size and the number of iterations, we show that a suitable choice of step-size and initialization-size can be computed from the data at the cost of one gradient-descent iteration. Consequently, the only tuneable hyper-parameter in practice is the number of iterations, which can be done using cross-validation as we have done in our experiments. With the provided choice of hyper-parameters, we show that Algorithm 1 attains statistical optimality after $\widetilde{O}(\sqrt{n})$ gradient-descent iterations, yielding a computationally sub-optimal algorithm. To circumvent this issue we propose a novel increasing step-sizes scheme (Algorithm 2) under which only $\widetilde{O}(1)$ iterations are required, resulting in a computationally optimal algorithm up to poly-logarithmic factors.

Our results can be improved in two different aspects. First, our constraints on the RIP parameter $\delta$ result in sub-optimal sample complexity with respect to the sparsity parameter $k$. Second, the RIP condition could potentially be replaced by the restricted eigenvalue (RE) condition which allows correlated designs. We expand on both of these points in Appendix I and provide empirical evidence suggesting that both inefficiencies are artifacts of our analysis and not inherent limitations of our algorithms.

**Acknowledgments**

Tomas Vaškevičius is supported by the EPSRC and MRC through the OxWaSP CDT programme (EP/L016710/1). Varun Kanade and Patrick Rebeschini are supported in part by the Alan Turing Institute under the EPSRC grant EP/N510129/1.

## Footnotes

[1]However, see Remark 1 for work concurrent to ours that achieves similar goals.

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
