[Supplementary Material]

## Appendix

The appendix is organized as follows.

In Appendix A we introduce the key ideas and intuition behind the proof of Theorem 1.

In Appendix B we go deeper into technical details and prove the main propositions used to prove Theorem 1.

In Appendix C we prove the lemmas stated in appendix A.

In Appendix D we prove Theorem 2.

In Appendix E we prove Theorem 3.

In Appendix F we derive the gradient descent updates used by our parametrization.

In Appendix G we compare our assumptions with the ones made in [30].

In Appendix H we compare our main result with a recent arXiv preprint [56], where Hadamard product reparametrization was used to induce sparsity implicitly.

In Appendix I we expand on the potential improvements discussed in Section 6.

In Appendix J we provide a table of notation.

## A   Proof of Theorem 1

This section is dedicated to providing a high level proof for Theorem 1. In Section A.1 we set up the notation and explain how we decompose our iterates into signal and error sequences. In Section A.2 we state and discuss the implications of the two key propositions allowing to prove our theorem. In Section A.3 we state some technical lemmas used in the proofs of the main theorem and its corollaries. In Section A.4 we prove Theorem 1. Finally in Section A.5 we prove the corollaries.

### A.1   Set Up and Intuition

Let $\mathbf{w}_t := \mathbf{w}_t^+ - \mathbf{w}_t^-$ where $\mathbf{w}_t^+ := \mathbf{u}_t \odot \mathbf{u}_t$ and $\mathbf{w}_t^- := \mathbf{v}_t \odot \mathbf{v}_t$. The gradient descent updates on $\mathbf{u}_t$ and $\mathbf{v}_t$ read as (see Appendix F for derivation)

$$\mathbf{u}_{t+1} = \mathbf{u}_t \odot \left( \mathbf{1} - 4\eta \left( \frac{1}{n} \mathbf{X}^\mathsf{T} (\mathbf{X}(\mathbf{w}_t - \mathbf{w}^\star) - \xi) \right) \right),$$

$$\mathbf{v}_{t+1} = \mathbf{v}_t \odot \left( \mathbf{1} + 4\eta \left( \frac{1}{n} \mathbf{X}^\mathsf{T} (\mathbf{X}(\mathbf{w}_t - \mathbf{w}^\star) - \xi) \right) \right).$$

Let $S^+$ denote the coordinates of $\mathbf{w}^\star$ such that $w_i^\star > 0$ and let $S^-$ denote the coordinates of $\mathbf{w}^\star$ such that $w_i^\star < 0$. So $S = S^+ \cup S^-$ and $S^+ \cap S^- = \emptyset$. Then define the following sequences

$$\mathbf{s}_t := \mathbf{1}_{S^+} \odot \mathbf{w}_t^+ - \mathbf{1}_{S^-} \odot \mathbf{w}_t^-,$$

$$\mathbf{e}_t := \mathbf{1}_{S^c} \odot \mathbf{w}_t + \mathbf{1}_{S^-} \odot \mathbf{w}_t^+ - \mathbf{1}_{S^+} \odot \mathbf{w}_t^-,$$

$$\mathbf{b}_t := \frac{1}{n} \mathbf{X}^\mathsf{T} \mathbf{X} \mathbf{e}_t - \frac{1}{n} \mathbf{X}^\mathsf{T} \xi, \tag{3}$$

$$\mathbf{p}_t := \left( \frac{1}{n} \mathbf{X}^\mathsf{T} \mathbf{X} - \mathbf{I} \right) (\mathbf{s}_t - \mathbf{w}^\star).$$

Having defined the sequences above we can now let $\alpha^2$ be the initialization size and rewrite the updates on $\mathbf{w}_t$, $\mathbf{w}_t^+$ and $\mathbf{w}_t^-$ in a more succinct way

$$\mathbf{w}_0^+ = \mathbf{w}_0^- = \alpha^2,$$

$$\mathbf{w}_t = \mathbf{w}_t^+ - \mathbf{w}_t^-$$

$$\mathbf{w}_{t+1}^+ = \mathbf{w}_t^+ \odot \left( \mathbf{1} - 4\eta \left( \mathbf{s}_t - \mathbf{w}^\star + \mathbf{p}_t + \mathbf{b}_t \right) \right)^2, \tag{4}$$

$$\mathbf{w}_{t+1}^- = \mathbf{w}_t^- \odot \left( \mathbf{1} + 4\eta \left( \mathbf{s}_t - \mathbf{w}^\star + \mathbf{p}_t + \mathbf{b}_t \right) \right)^2.$$

We will now explain the roles played by each sequence defined in equation (3).

1. The sequence $(\mathbf{s}_t)_{t\geq 0}$ represents the signal that we have fit by iteration $t$. In the noiseless setting, $\mathbf{s}_t$ would converge to $\mathbf{w}^\star$. We remark that $\mathbf{w}_t^+$ is responsible for fitting the positive components of $\mathbf{w}^\star$ while $\mathbf{w}_t^-$ is responsible for fitting the negative components of $\mathbf{w}^\star$. If we had the knowledge of $S^+$ and $S^-$ before starting our algorithm, we would set $\mathbf{w}_0$ to $\mathbf{s}_0$.

2. The sequence $(\mathbf{e}_t)_{t\geq 0}$ represents the error sequence. It has three components: $\mathbf{1}_{S^c} \odot \mathbf{w}_t$, $\mathbf{1}_{S^-} \odot \mathbf{w}_t^+$ and $\mathbf{1}_{S^+} \odot \mathbf{w}_t^-$ which represent the errors of our estimator arising due to not having the knowledge of $S^c$, $S^+$ and $S^-$ respectively. For example, if we knew that $\mathbf{w}^\star \succcurlyeq 0$ we could instead use the parametrization $\mathbf{w}_0 = \mathbf{u}_0 \odot \mathbf{u}_0 = \mathbf{w}_0^+$ while if we knew that $\mathbf{w}^\star \preccurlyeq 0$ then we would use the parametrization $\mathbf{w}_0 = -\mathbf{v}_0 \odot \mathbf{v}_0 = -\mathbf{w}_0^-$.

   A key property of our main results is that we stop running gradient descent before $\|\mathbf{e}_t\|_\infty$ exceeds some function of initialization size. This allows us to recover the coordinates from the true support $S$ that are sufficiently above the noise level while keeping the coordinates outside the true support arbitrarily close to 0.

3. We will think of the sequence $(\mathbf{b}_t)_{t\geq 0}$ as a sequence of bounded perturbations to our gradient descent updates. These perturbations come from two different sources. The first one is the term $\frac{1}{n}\mathbf{X}^\mathsf{T}\xi$ which arises due to the noise on the labels. Hence this part of error is never greater than $\left\|\frac{1}{n}\mathbf{X}^\mathsf{T}\xi\right\|_\infty$ and is hence bounded with high probability in the case of subGaussian noise. The second source of error is $\frac{1}{n}\mathbf{X}^\mathsf{T}\mathbf{X}\mathbf{e}_t$ and it comes from the error sequence $(\mathbf{e}_t)_{t\geq 0}$ being non-zero. Even though this term is in principle can be unbounded, as remarked in the second point above, we will always stop running gradient descent while $\|\mathbf{e}_t\|_\infty$ remains close enough to 0. In particular, this allows to treat $\frac{1}{n}\mathbf{X}^\mathsf{T}\mathbf{X}\mathbf{e}_t$ as a bounded error term .

4. We will refer to the final error sequence $(\mathbf{p}_t)_{t\geq 0}$ as a sequence of errors proportional to convergence distance. An intuitive explanation of the restricted isometry property is that $\frac{1}{n}\mathbf{X}^\mathsf{T}\mathbf{X} \approx \mathbf{I}$ for sparse vectors. The extent to which this approximation is exact is controlled by the RIP parameter $\delta$. Hence the sequence $(\mathbf{p}_t)_{t\geq 0}$ represents the error arising due to $\frac{1}{\sqrt{n}}\mathbf{X}$ not being an exact isometry for sparse vectors in a sense that $\delta \neq 0$. If we require that $\delta \leq \gamma/\sqrt{k}$ for some $\gamma > 0$ then as we shall see in section A.3 we can upper bound $\|\mathbf{p}_t\|_\infty$ as
$$\|\mathbf{p}_t\|_\infty \leq \delta \|\mathbf{s}_t - \mathbf{w}^\star\|_2 \leq \gamma \|\mathbf{s}_t - \mathbf{w}^\star\|_\infty .$$
Since this is the only worst-case control we have on $(\mathbf{p}_t)_{t\geq 0}$ one may immediately see the most challenging part of our analysis. For small $t$ we have $\mathbf{s}_t \approx 0$ and hence in the worst case $\|\mathbf{p}_t\|_\infty \approx \gamma \|\mathbf{w}^\star\|_\infty$. Since $\|\mathbf{w}^\star\|_\infty$ can be arbitrarily large, we can hence see that while $t$ is small it is possible for some elements of $(\mathbf{e}_t)_{t\geq 0}$ to grow at a very fast rate, while some of the signal terms in the sequence $\mathbf{s}_t$ can actually shrink, for example, if $\gamma \|\mathbf{w}^\star\|_\infty > |w_i^\star|$ for some $i \in S$. We address this difficulty in Section B.3.

One final thing to discuss regarding our iterates $\mathbf{w}_t$ is how to initialize $\mathbf{w}_0$. Having the point two above in mind, we will always want $\|\mathbf{e}_t\|_\infty$ to be as small as possible. Hence we should initialize the sequences $(\mathbf{u}_t)_{t\geq 0}$ and $(\mathbf{v}_t)_{t\geq 0}$ as close to 0 as possible. Note, however, that due to the multiplicative nature of gradient descent updates using our parametrization, we cannot set $\mathbf{u}_0 = \mathbf{v}_0 = 0$ since this is a saddle point for our optimization objective function. We will hence set $\mathbf{u}_0 = \mathbf{v}_0 = \alpha$ for some small enough positive real number $\alpha$.

Appendix B is dedicated to understanding the behavior of the updates given in equation (4). In appendix B.1 we analyze behavior of $(\mathbf{w}_t^+)_{t\geq 0}$ assuming that $\mathbf{w}_t^- = 0$, $\mathbf{p}_t = 0$ and $\mathbf{b}_t = 0$. In appendix B.2 we show how to handle the bounded errors sequence $(\mathbf{b}_t)_{t\geq 0}$ and in appendix B.3 we show how to deal with the errors proportional to convergence distance $(\mathbf{p}_t)_{t\geq 0}$. Finally, in appendix B.4 we show how to deal with sequences $(\mathbf{w}_t^+)_{t\geq 0}$ and $(\mathbf{w}_t^-)_{t\geq 0}$ simultaneously.

## A.2  The Key Propositions

In this section we state the key propositions appearing in the proof of Theorem 1 and discuss their implications.

Proposition 1 is the core of our proofs. It allows to ignore the error sequence $(\mathbf{p}_t)_{t\geq 0}$ as long as the RIP constant $\delta$ is small enough. That is, suppose that $\|\mathbf{b}_t\|_\infty \lesssim \zeta$ for some $\zeta > 0$. Proposition 1 states that if $\delta \lesssim 1/\sqrt{k}(\log \frac{w^\star_{\max}}{\zeta} \vee 1)$ then it is possible to fit the signal sequence $(\mathbf{s}_t)_{t\geq 0}$ to $\mathbf{w}^\star$ up to precision proportional to $\zeta$ while keeping the error sequence $(\mathbf{e}_t)_{t\geq 0}$ arbitrarily small. See appendix B.5 for proof.

**Proposition 1.** *Consider the setting of updates given in equations* (3) *and* (4). *Fix any* $0 < \zeta \leq w^\star_{\max}$ *and let* $\gamma = \frac{C_\gamma}{\left\lceil \log_2 \frac{w^\star_{\max}}{\zeta} \right\rceil}$ *where* $C_\gamma$ *is some small enough absolute constant. Suppose the error sequences* $(\mathbf{b}_t)_{t\geq 0}$ *and* $(\mathbf{p}_t)_{t\geq 0}$ *for any* $t \geq 0$ *satisfy the following:*

$$\|\mathbf{b}_t\|_\infty \leq C_b \zeta - \alpha,$$
$$\|\mathbf{p}_t\|_\infty \leq \gamma \|\mathbf{s}_t - \mathbf{w}^\star\|_\infty ,$$

*where* $C_b$ *is some small enough absolute constant. If the step size satisfies* $\eta \leq \frac{5}{96 w^\star_{\max}}$ *and the initialization satisfies* $\alpha \leq 1 \wedge \frac{\zeta}{3(w^\star_{\max})^2} \wedge \frac{1}{2}\sqrt{w^\star_{\min}}$ *Then, for some* $T = O\left(\frac{1}{\eta\zeta}\log\frac{1}{\alpha}\right)$ *and any* $0 \leq t \leq T$ *we have*

$$\|\mathbf{s}_T - \mathbf{w}^\star\|_\infty \leq \zeta,$$
$$\|\mathbf{e}_t\|_\infty \leq \alpha.$$

The proof of Theorem 1 in the hard regime when $w^\star_{\min} \lesssim \left\|\frac{1}{n}\mathbf{X}^\mathsf{T}\xi\right\|_\infty \vee \varepsilon$ is then just a simple application of the above theorem with $\zeta = \frac{2}{C_b}(\left\|\frac{1}{n}\mathbf{X}^\mathsf{T}\xi\right\|_\infty \vee \varepsilon)$ where the absolute constant $C_b$ needs to satisfy the conditions of the above proposition.

On the other hand, if $w^\star_{\min} \gtrsim \left\|\frac{1}{n}\mathbf{X}^\mathsf{T}\xi\right\|_\infty \vee \varepsilon$ which happens as soon as we choose small enough $\varepsilon$ and when we get enough data points $n$, we can apply Proposition 1 with $\zeta = \frac{1}{5}w^\star_{\min}$. Then, after $O(\frac{1}{\eta w^\star_{\min}}\log\frac{1}{\alpha})$ iterations we can keep $\|\mathbf{e}_t\|_\infty$ below $\alpha$ while $\|\mathbf{s}_t - \mathbf{w}^\star\|_\infty \leq \frac{1}{5}w^\star_{\min}$. From this point onward, the convergence of the signal sequence $(\mathbf{s}_t)_{t\geq 0}$ does not depend on $\alpha$ anymore while the error term is smaller than $\alpha$. We can hence fit the signal sequence to $\mathbf{w}^\star$ up to precision $\left\|\frac{1}{n}\mathbf{X}^\mathsf{T}\xi \odot \mathbf{1}_S\right\|_\infty \vee \varepsilon$ while keeping $\|\mathbf{e}_t\|_\infty$ arbitrarily small. This idea is formalized in the following proposition.

**Proposition 2.** *Consider the setting of updates given in equations* (3) *and* (4). *Fix any* $\varepsilon > 0$ *and suppose that the error sequences* $(\mathbf{b}_t)_{t\geq 0}$ *and* $(\mathbf{p}_t)_{t\geq 0}$ *for any* $t \geq 0$ *satisfy*

$$\|\mathbf{b}_t \odot \mathbf{1}_i\|_\infty \leq B_i \leq \frac{1}{10}w^\star_{\min},$$

$$\|\mathbf{p}_t\|_\infty \leq \frac{1}{20}\|\mathbf{s}_t - \mathbf{w}^\star\|_\infty .$$

*Suppose that*

$$\|\mathbf{s}_0 - \mathbf{w}^\star\|_\infty \leq \frac{1}{5}w^\star_{\min}.$$

*Let the step size satisfy* $\eta \leq \frac{5}{96 w^\star_{\max}}$. *Then for all* $t \geq 0$

$$\|\mathbf{s}_t - \mathbf{w}^\star\|_\infty \leq \frac{1}{5}w^\star_{\min}$$

*and for any* $t \geq \frac{45}{32\eta w^\star_{\min}}\log\frac{w^\star_{\min}}{\varepsilon}$ *and for any* $i \in S$ *we have*

$$|s_{t,i} - w^\star_i| \lesssim \delta\sqrt{k}\max_{j \in S} B_j \vee B_i \vee \varepsilon.$$

### A.3  Technical Lemmas

In this section we state some technical lemmas which will be used to prove Theorem 1 and its corollaries. Proofs for all of the lemmas stated in this section can be found in Appendix C.

We begin with Lemma A.1 which allows to upper-bound the error sequence $(\mathbf{e}_t)_{t\geq 0}$ in terms of sequences $(\mathbf{b}_t)_{t\geq 0}$ and $(\mathbf{p}_t)_{t\geq 0}$.

**Lemma A.1.** *Consider the setting of updates given in equations(3) and (4). Suppose that $\|\mathbf{e}_0\|_\infty \leq \frac{1}{4} w_{\min}^\star$ and that there exists some $B \in \mathbb{R}$ such that for all $t$ we have $\|\mathbf{b}_t\|_\infty + \|\mathbf{p}_t\|_\infty \leq B$. Then, if $\eta \leq \frac{1}{12(w_{\max}^\star + B)}$ for any $t \geq 0$ we have*

$$\|\mathbf{e}_t\|_\infty \leq \|\mathbf{e}_0\|_\infty \prod_{i=0}^{t-1} (1 + 4\eta(\|\mathbf{b}_i\|_\infty + \|\mathbf{p}_i\|_\infty))^2.$$

Once we have an upper-bound on $\|\mathbf{p}_t\|_\infty + \|\mathbf{b}_t\|_\infty$ we can apply Lemma A.2 to control the size of $\|\mathbf{e}_t\|_\infty$. This happens, for example, in the easy setting when $w_{\min}^\star \gtrsim \left\|\frac{1}{n}\mathbf{X}^\mathsf{T}\xi\right\|_\infty \vee \varepsilon$ where after the application of Proposition 1 we have $\|\mathbf{p}_t\|_\infty + \|\mathbf{b}_t\|_\infty \lesssim w_{\min}^\star$.

**Lemma A.2.** *Let $(b_t)_{t \geq 0}$ be a sequence such that for any $t \geq 0$ we have $|b_t| \leq B$ for some $B > 0$. Let the step size $\eta$ satisfy $\eta \leq \frac{1}{8B}$ and consider a one-dimensional sequence $(x_t)_{t \geq 0}$ given by*

$$0 < x_0 < 1,$$
$$x_{t+1} = x_t(1 + 4\eta b_t)^2.$$

*Then for any $t \leq \frac{1}{32\eta B} \log \frac{1}{x_0^2}$ we have*

$$x_t \leq \sqrt{x_0}.$$

We now introduce the following two lemmas related to the restricted isometry property. Lemma A.3 allows to control the $\ell_\infty$ norm of the sequence $(\mathbf{p}_t)_{t \geq 0}$. Lemma A.4 allows to control the $\ell_\infty$ norm of the term $\frac{1}{n}\mathbf{X}^\mathsf{T}\mathbf{X}\mathbf{e}_t$ arising in the bounded errors sequence $(\mathbf{b}_t)_{t \geq 0}$.

**Lemma A.3.** *Suppose that $\frac{1}{\sqrt{n}}\mathbf{X}$ is a $n \times d$ matrix satisfying the $(k+1, \delta)$-RIP. If $\mathbf{z} \in \mathbb{R}^d$ is a $k$-sparse vector then*

$$\left\| \left( \frac{1}{n}\mathbf{X}^\mathsf{T}\mathbf{X} - I \right) \mathbf{z} \right\|_\infty \leq \sqrt{k}\delta \|\mathbf{z}\|_\infty.$$

**Lemma A.4.** *Suppose that $\frac{1}{\sqrt{n}}\mathbf{X}$ is a $n \times d$ matrix satisfying the $(1, \delta)$-RIP with $0 \leq \delta \leq 1$ and let $\mathbf{X}_i$ be the $i^{th}$ column of $\mathbf{X}$. Then*

$$\max_i \left\| \frac{1}{\sqrt{n}}\mathbf{X}_i \right\|_2 \leq \sqrt{2}$$

*and for any vector $\mathbf{z} \in \mathbb{R}^d$ we have*

$$\left\| \frac{1}{n}\mathbf{X}^\mathsf{T}\mathbf{X}\mathbf{z} \right\|_\infty \leq 2d \|\mathbf{z}\|_\infty.$$

Finally, we introduce a lemma upper-bounding the maximum noise term $\left\|\frac{1}{n}\mathbf{X}^\mathsf{T}\xi\right\|_\infty$ when $\xi$ is subGaussian with independent entries and the design matrix $\mathbf{X}$ is treated as fixed.

**Lemma A.5.** *Let $\frac{1}{\sqrt{n}}\mathbf{X}$ be a $n \times d$ matrix such that the $\ell_2$ norms of its columns are bounded by some absolute constant $C$. Let $\xi \in \mathbb{R}^n$ be a vector of independent $\sigma^2$-subGaussian random variables. Then, with probability at least $1 - \frac{1}{8d^3}$*

$$\left\| \frac{1}{n}\mathbf{X}^\mathsf{T}\xi \right\|_\infty \lesssim \sqrt{\frac{\sigma^2 \log d}{n}}.$$

### A.4    Proof of Theorem 1

Let $C_b$ and $C_\gamma$ be small enough absolute positive constants that satisfy conditions of Proposition 1.

Let

$$\zeta := \frac{1}{5} w_{\min}^\star \vee \frac{2}{C_b} \left\| \frac{1}{n}\mathbf{X}^\mathsf{T}\xi \right\|_\infty \vee \frac{2}{C_b}\varepsilon.$$

and suppose that

$$\delta \leq \frac{C_\gamma}{\sqrt{k}\left( \log_2 \frac{w_{\max}^\star}{\zeta} + 1 \right)}.$$

Setting

$$\alpha \leq 1 \wedge \frac{\varepsilon^2}{(2d+1)^2} \wedge \frac{\varepsilon}{w_{\max}^\star} \wedge \frac{\zeta}{3(w_{\max}^\star)^2} \wedge \frac{1}{2}\sqrt{w_{\min}^\star}$$

we satisfy pre-conditions of Proposition 1. Also, by Lemma A.4 as long as $\|\mathbf{e}_t\|_\infty \leq \sqrt{\alpha}$ we have

$$\left\| \frac{1}{n}\mathbf{X}^\mathsf{T}\mathbf{X}\mathbf{e}_t \right\|_\infty + \alpha \leq (2d+1)\sqrt{\alpha} \leq \varepsilon.$$

It follows that as long as $\|\mathbf{e}_t\|_\infty \leq \sqrt{\alpha}$ we can upper bound $\|\mathbf{b}_t\|_\infty + \alpha$ as follows:

$$\|\mathbf{b}_t\|_\infty + \alpha \leq \left\| \frac{1}{n}\mathbf{X}^\mathsf{T}\xi \right\|_\infty + \varepsilon \leq C_b \cdot \frac{2}{C_b}\left( \left\| \frac{1}{n}\mathbf{X}^\mathsf{T}\xi \right\|_\infty \vee \varepsilon \right) \leq C_b\zeta.$$

By Lemma A.3 we also have

$$\|\mathbf{p}_t\|_\infty \leq \frac{C_\gamma}{\left\lceil \log_2 \frac{w_{\max}^\star}{\zeta} \right\rceil} \|\mathbf{s}_t - \mathbf{w}^\star\|_\infty .$$

and so both sequences $(\mathbf{b}_t)_{t\geq 0}$ and $(\mathbf{p}_t)_{t\geq 0}$ satisfy the assumptions of Proposition 1 conditionally on $\|\mathbf{e}_t\|_\infty$ staying below $\sqrt{\alpha}$. If $\zeta \geq w_{\max}^\star$ then the statement of our theorem already holds at $t = 0$ and we are done. Otherwise, applying Proposition 1 we have after

$$T = O\left( \frac{1}{\eta\zeta} \log \frac{1}{\alpha} \right)$$

iterations

$$\|\mathbf{s}_T - \mathbf{w}^\star\|_\infty \leq \zeta$$
$$\|\mathbf{e}_T\|_\infty \leq \alpha.$$

If $\frac{1}{5}w_{\min}^\star \leq \frac{2}{C_b}\left\| \frac{1}{n}\mathbf{X}^\mathsf{T}\xi \right\|_\infty \vee \frac{2}{C_b}\varepsilon$ then we are in what we refer to as the hard regime and we are done.

On the other hand, suppose that $\frac{1}{5}w_{\min}^\star > \frac{2}{C_b}\left\| \frac{1}{n}\mathbf{X}^\mathsf{T}\xi \right\|_\infty \vee \frac{2}{C_b}\varepsilon$ so that we are working in the easy regime and $\zeta = \frac{1}{5}w_{\min}^\star$.

Conditionally on $\|\mathbf{e}_t\|_\infty \leq \sqrt{\alpha}$, $\|\mathbf{p}_t\|_\infty$ stays below $C_\gamma \cdot \frac{1}{5}w_{\min}^\star$ by Proposition 2. Hence,

$$\|\mathbf{b}_t\|_\infty + \|\mathbf{p}_t\|_\infty \leq (C_b + C_\gamma) \cdot \frac{1}{5}w_{\min}^\star.$$

Applying Lemmas A.1 and A.2 we can maintain that $\|\mathbf{e}_t\|_\infty \leq \sqrt{\alpha}$ for at least another $\frac{5}{16(C_b+C_\gamma)\eta w_{\min}^\star} \log \frac{1}{\alpha}$ iterations after an application of Proposition 1. Crucially, with a small enough $\alpha$ we can maintain the above property for as long as we want and in our case here we need $\alpha \leq \varepsilon/w_{\max}^\star$.

Choosing small enough $C_b$ and $C_\gamma$ so that $C_b + C_\gamma \leq \frac{2}{9}$ and $C_\gamma \leq \frac{1}{20}$ and applying Proposition 1 we have after

$$T' := T + \frac{45}{32\eta w_{\min}^\star} \log \frac{w_{\min}^\star}{\varepsilon} \leq T + \frac{5}{16(C_b+C_\gamma)\eta w_{\min}^\star} \log \frac{1}{\alpha}$$

iterations

$$\|\mathbf{e}_{T'}\|_\infty \leq \sqrt{\alpha}$$

and for any $i \in S$

$$|s_{T',i} - w_i^\star| \lesssim \sqrt{k}\delta \left\| \frac{1}{n}\mathbf{X}^\mathsf{T}\xi \odot \mathbf{1}_S \right\|_\infty \vee \left| \left( \frac{1}{n}\mathbf{X}^\mathsf{T}\xi \right)_i \right| \vee \varepsilon.$$

Finally, noting that for all $t \leq T'$ we have

$$|w_{t,i} - w^\star| \leq |s_{t,i} - w_i^\star| + |e_{t,i}| \leq |s_{t,i} - w_i^\star| + \sqrt{\alpha} \leq |s_{t,i} - w_i^\star| + \varepsilon$$

our result follows.

## A.5 Proofs of Corollaries

*Proof of Corollary 1.* Since $\xi = 0$ the bound in Theorem 1 directly reduces to

$$\|\mathbf{w}_t - \mathbf{w}^\star\|_2^2 \lesssim \sum_{i \in S} \varepsilon^2 + \sum_{i \notin S} \alpha \leq k\varepsilon^2 + (d-k)\frac{\varepsilon^2}{(2d+1)^2} \lesssim k\varepsilon^2.$$

$\square$

*Proof of Corollary 2.* By Lemma A.4 and the proof of Lemma A.5 with probability at least $1 - 1/(8d^3)$ we have

$$\left\|\frac{1}{n}\mathbf{X}^\mathsf{T}\xi\right\|_\infty \leq 4\frac{\sqrt{2\sigma^2 \log(2d)}}{\sqrt{n}}.$$

Hence, letting $\varepsilon = 4\frac{\sqrt{2\sigma^2 \log(2d)}}{\sqrt{n}}$, Theorem 1 implies with probability at least $1 - 1/(8d^3)$

$$\|\mathbf{w}_t - \mathbf{w}^\star\|_2^2 \lesssim \sum_{i \in S} \varepsilon^2 + \sum_{i \notin S} \alpha \leq k\varepsilon^2 + (d-k)\frac{\varepsilon^2}{(2d+1)^2} \lesssim \frac{k\sigma^2 \log d}{n}.$$

$\square$

*Proof of Corollary 3.* We use the same argument as in proof of Corollary 3 with the term $\|\mathbf{X}^\mathsf{T}\xi\|_\infty/n$ replaced with $\sqrt{k}\delta\|\mathbf{X}^\mathsf{T}\xi \odot \mathbf{1}_S\|_\infty/n$. Since $\sqrt{k}\delta \lesssim 1$ an identical result holds with $d$ replaced with $k$.

$\square$

# B   Understanding Multiplicative Update Sequences

In this section of the appendix, we provide technical lemmas to understand the behavior of multiplicative updates sequences. We then prove Propositions 1 and 2.

## B.1   Basic Lemmas

In this section we analyze one-dimensional sequences with positive target corresponding to gradient descent updates without any perturbations. That is, this section corresponds to parametrization $\mathbf{w}_t = \mathbf{u}_t \odot \mathbf{u}_t$ and gradient descent updates under assumption that $\frac{1}{n}\mathbf{X}^\mathsf{T}\mathbf{X} = \mathbf{I}$ and ignoring the error sequences $(\mathbf{b}_t)_{t \geq 0}$ and $(\mathbf{p}_t)_{t \geq 0}$ given in equation (3) completely. We will hence look at one-dimensional sequences of the form

$$
\begin{aligned}
&0 < x_0 = \alpha^2 < x^\star \\
&x_{t+1} = x_t(1 - 4\eta(x_t - x^\star))^2.
\end{aligned}
\tag{5}
$$

Recall the definition of gradient descent updates given in equations (3) and (4) and let $\mathbf{v}_t = 0$ for all $t$. Ignoring the effects of the sequence $(\mathbf{p}_t)_{t \geq 0}$ and the term $\frac{1}{n}\mathbf{X}^\mathsf{T}\mathbf{X}\mathbf{e}_t$ one can immediately see that $\|\mathbf{1}_{S^c} \odot \mathbf{w}_t\|_\infty$ grows at most as fast as the sequence $(x_t)_{t \geq 0}$ given in equation (5) with $x^\star = \left\|\frac{1}{n}\mathbf{X}^\mathsf{T}\xi\right\|_\infty$. Surely, for any $i \in S$ such that $0 < w_i^\star < \left\|\frac{1}{n}\mathbf{X}^\mathsf{T}\xi\right\|_\infty$ we cannot fit the $i-th$ component of $w^\star$ without fitting any of the noise variables $\mathbf{1}_{S^c} \odot w_t$. On the other hand, for any $i \in S$ such that $w_i^\star \gg \left\|\frac{1}{n}\mathbf{X}^\mathsf{T}\xi\right\|_\infty$ can fit the sequence $(x_t)_{t \geq 0}$ with $x^\star = w_i^\star$ while keeping all of the noise variables arbitrarily small, as we shall see in this section.

We can hence formulate a precise question that we answer in this section. Consider two sequences $(x_t)_{t \geq 0}$ and $(y_t)_{t \geq 0}$ with updates as in equation (5) with targets $x^\star$ and $y^\star$ respectively. One should think of the sequence $(y_t)_{t \geq 0}$ as a sequence fitting the noise, so that $y^\star = \left\|\frac{1}{n}\mathbf{X}^\mathsf{T}\xi\right\|_\infty$. Let $T_\alpha^y$ be the smallest $t \geq 0$ such that $y_t \geq \alpha$. On the other hand, one should think of sequence $(x_t)_{t \geq 0}$ as a sequence fitting the signal. Let $T_{x^\star - \varepsilon}^x$ be the smallest $t$ such that $x_t \geq x^\star - \varepsilon$. Since we want to fit the sequence $(x_t)_{t \geq 0}$ to $x^\star$ within $\varepsilon$ error before $(y_t)_{t \geq 0}$ exceeds $\alpha$ we want $T_{x^\star - \varepsilon}^x \leq T_\alpha^y$. This can only hold if the variables $x^\star, y^\star, \alpha$ and $\varepsilon$ satisfy certain conditions. For instance, decreasing $\varepsilon$ will increase $T_{x^\star - \varepsilon}^x$ without changing $T_\alpha^y$. Also, if $x^\star < y^\star$ then satisfying $T_{x^\star - \varepsilon}^x \leq T_\alpha^y$ is impossible for sufficiently small $\varepsilon$. However, as we shall see in this section, if $x^\star$ is sufficiently bigger than $y^\star$

Figure 5: The blue and red lines represent the signal sequence $(x_t)_{t \geq 0}$ and the noise sequence $(y_t)_{t \geq 0}$ plotted on $\log$ scale. The vertical blue and red dashed lines show the hitting times $T^x_{x^\star - \varepsilon}$ and $T^y_\alpha$ so that we want the blue vertical line to appear on the left side of the red vertical line. Both plots use the same values of $x^\star$, $y^\star$ and $\varepsilon$. However, the plot on the left is plotted with $\alpha = 10^{-2}$ and the plot on the right is plotted with $\alpha = 10^{-8}$. This shows the effect of decreasing initialization size; both vertical lines are pushed to the right, but the red vertical line is pushed at a faster pace.

then for any $\varepsilon > 0$ one can choose a small enough $\alpha$ such that $T^x_{x^\star - \varepsilon} \leq T^y_\alpha$. To see this intuitively, note that if we ignore what happens when $x_t$ gets close to $x^\star$, the sequence $(x_t)_{t \geq 0}$ behaves as an exponential function $t \mapsto \alpha^2 (1 + 4\eta x^\star)^{2t}$ while the sequence $y^\star$ behaves as $t \mapsto \alpha^2 (1 + 4\eta y^\star)^{2t}$. Since exponential function is very sensitive to its base, we can make the gap between $\alpha^2 (1 + 4\eta x^\star)^{2t}$ and $\alpha^2 (1 + 4\eta y^\star)^{2t}$ as big as we want by decreasing $\alpha$ and increasing $t$. This intuition is depicted in Figure 5.

With the above discussion in mind, in this section we will quantitatively formalize under what conditions on $x^\star, y^\star, \alpha$ and $\varepsilon$ the inequality $T^x_{x^\star - \varepsilon} \leq T^y_\alpha$ hold. We begin by showing that for small enough step sizes, multiplicative update sequences given in equation (5) behave monotonically.

**Lemma B.6** (Iterates behave monotonically). *Let $\eta > 0$ be the step size and suppose that updates are given by*

$$x_{t+1} = x_t (1 - 4\eta(x_t - x^\star))^2.$$

*Then the following holds*

1. *If $0 < x_0 \leq x^\star$ and $\eta \leq \frac{1}{8x^\star}$ then for any $t > 0$ we have $x_0 \leq x_{t-1} \leq x_t \leq x^\star$.*

2. *If $x^\star \leq x_0 \leq \frac{3}{2} x^\star$ and $\eta \leq \frac{1}{12x^\star}$ then for any $t > 0$ we have $x^\star \leq x_t \leq x_{t-1} \leq x_0$.*

*Proof.* Note that if $x_0 \leq x_t \leq x^\star$ then $x_t - x^\star \leq 0$ and hence $x_{t+1} \geq x_t$. Thus for the first part it is enough to show that for all $t \geq 0$ we have $x_t \leq x^\star$.

Assume for a contradiction that exists $t$ such that

$$x_0 \leq x_t \leq x^\star,$$
$$x_{t+1} > x^\star.$$

Plugging in the update rule for $x_{t+1}$ we can rewrite the above as

$$x_t \leq x^\star$$
$$< x_t(1 - 4\eta(x_t - x^\star))^2$$
$$\leq x_t \left(1 + \frac{1}{2} - \frac{x_t}{2x^\star}\right)^2$$

Letting $\lambda := \frac{x_t}{x^\star}$ we then have by our assumption above $0 < \lambda \leq 1$. The above inequality then gives us

$$\sqrt{\frac{1}{\lambda}} < 3/2 - \frac{1}{2}\lambda$$

And hence for $0 < \lambda \leq 1$ we have $f(\lambda) := \sqrt{\frac{1}{\lambda} + \frac{1}{2}\lambda} < 3/2$. Since for $0 < \lambda < 1$ we also have $f'(\lambda) = \frac{1}{2}(1 - \frac{1}{\lambda^{3/2}}) < 0$ and so $f(\lambda) \geq f(1) = 3/2$. This gives us the desired contradiction and concludes our proof for the first part.

We will now prove the send part. Similarly to the first part, we just need to show that for all $t \geq 0$ we have $x_t \geq x^\star$. Suppose that $\frac{3}{2}x^\star \geq x_t \geq x^\star$ and hence we can write $x_t = x^\star(1 + \gamma)$ for some $\gamma \in [0, \frac{1}{2}]$. Then we have

$$
\begin{aligned}
x_{t+1} &= (1+\gamma)x^\star(1 - 4\eta\gamma x^\star)^2 \\
&\geq (1+\gamma)x^\star(1 - \frac{1}{3}\gamma)^2.
\end{aligned}
$$

One may verify that the polynomial $(1+\gamma)(1 - \frac{1}{3}\gamma)^2$ is no smaller than one for $0 \leq \gamma \leq \frac{1}{2}$ which finishes the second part of our proof. $\qquad\square$

While the above lemma tells us that for small enough step sizes the iterates are monotonic and bounded, the following two lemmas tell us that we are converging to the target exponentially fast. We first look at the behavior near convergence.

**Lemma B.7** (Iterates behaviour near convergence). *Consider the setting of Lemma B.6. Let $x^\star > 0$ and and suppose that $|x_0 - x^\star| \leq \frac{1}{2}x^\star$. Then the following holds.*

1. *If $0 < x_0 \leq x^\star$ and $\eta \leq \frac{1}{8x^\star}$ then for any $t \geq \frac{1}{4\eta x^\star}$ we have*

$$
0 \leq x^\star - x_t \leq \frac{1}{2}|x_0 - x^\star|.
$$

2. *If $x^\star \leq x_0 \leq \frac{3}{2}x^\star$ and $\eta \leq \frac{1}{12x^\star}$ then for any $t \geq \frac{1}{8\eta x^\star}$ we have*

$$
0 \leq x_t - x^\star \leq \frac{1}{2}|x_0 - x^\star|.
$$

*Proof.* Let us write $|x_0 - x^\star| = \gamma x^\star$ where $\gamma \in [0, \frac{1}{2}]$.

For the first part we have $x_0 = (1 - \gamma)x^\star$. Note that while $x_t \leq (1 - \frac{\gamma}{2})x^\star$ we have $x_{t+1} \geq x_t(1 + 4\eta\frac{\gamma}{2}x^\star)$. Recall that by the Lemma B.6 for all $t \geq 0$ we have $x_t \leq x^\star$. Hence to find $t$ such that $x^\star \geq x_t \geq (1 - \frac{\gamma}{2})x^\star$ it is enough to find a big enough $t$ satisfying the following inequality

$$
x_0(1 + 2\eta\gamma x^\star)^{2t} \geq \left(1 - \frac{\gamma}{2}\right)x^\star.
$$

Noting that for $x > 0$ ant $t \geq 1$ we have $(1 + x)^t \geq 1 + tx$ we have

$$
x_0(1 + 2\eta\gamma x^\star)^{2t} \geq x_0(1 + 4\eta\gamma x^\star t)
$$

and hence it is enough to find a big enough $t$ satisfying

$$
\begin{aligned}
x_0(1 + 4\eta\gamma x^\star t) &\geq \left(1 - \frac{\gamma}{2}\right)x^\star \\
\iff 4\eta\gamma x^\star t &\geq \frac{\left(1 - \frac{\gamma}{2}\right)x^\star - x_0}{x_0} \\
\iff 4\eta\gamma x^\star t &\geq \frac{\gamma}{2(1 - \gamma)} \\
\iff t &\geq \frac{1}{8\eta x^\star}\frac{1}{(1 - \gamma)}
\end{aligned}
$$

and since $\gamma \in [0, \frac{1}{2}]$ choosing $t \geq \frac{1}{4\eta x^\star}$ is enough.

To deal with the second part, now let us write $x_0 = x^\star(1 + \gamma)$. We will use a similar approach to the one used in the first part. If for some $x_t$ we have $x_t \leq (1 + \frac{\gamma}{2})x^\star$ by Lemma B.6 we would be done.

If $x_t > x^\star(1 + \frac{\gamma}{2})$ we have $x_{t+1} \leq x_t(1 - 4\eta\frac{\gamma}{2}x^\star)^2$. This can happen for at most $\frac{1}{8\eta x^\star}$ iterations, since

$$x_0(1 - 2\eta\gamma x^\star)^{2t} \leq x^\star(1 + \frac{\gamma}{2})$$

$$\Longleftrightarrow 2t\log(1 - 2\eta\gamma x^\star) \leq \log\frac{x^\star(1 + \frac{\gamma}{2})}{x^0}$$

$$\Longleftrightarrow t \geq \frac{1}{2}\frac{\log\frac{x^\star(1 + \frac{\gamma}{2})}{x^0}}{\log(1 - 2\eta\gamma x^\star)}.$$

We can deal with the term on the right hand side by noting that

$$\frac{1}{2}\frac{\log\frac{x^\star(1 + \frac{\gamma}{2})}{x^0}}{\log(1 - 2\eta\gamma x^\star)} = \frac{1}{2}\frac{\log\frac{1 + \frac{\gamma}{2}}{1 + \gamma}}{\log(1 - 2\eta\gamma x^\star)}$$

$$\leq \frac{1}{2}\frac{\left(\frac{1 + \frac{\gamma}{2}}{1 + \gamma} - 1\right)/\left(\frac{1 + \frac{\gamma}{2}}{1 + \gamma}\right)}{-2\eta\gamma x^\star}$$

$$= \frac{1}{2}\frac{-\frac{\gamma}{2}/\left(1 + \frac{\gamma}{2}\right)}{-2\eta\gamma x^\star}$$

$$\leq \frac{1}{8\eta x^\star}.$$

where in the second line we have used $\log x \leq x - 1$ and $\log x \geq \frac{x-1}{x}$. Note, however, that in the above inequalities both logarithms are negative, which is why the inequality signs are reversed. □

**Lemma B.8** (Iterates approach target exponentially fast). *Consider the setting of updates as in Lemma B.6 and fix any $\varepsilon > 0$.*

1. *If $\varepsilon < |x^\star - x_0| \leq \frac{1}{2}x^\star$ and $\eta \leq \frac{1}{12x^\star}$ then for any $t \geq \frac{3}{8\eta x^\star}\log\frac{|x^\star - x_0|}{\varepsilon}$ we have*

$$|x^\star - x_t| \leq \varepsilon.$$

2. *If $0 < x_0 \leq \frac{1}{2}x^\star$ and $\eta \leq \frac{1}{8x^\star}$ then for any $t \geq \frac{3}{8\eta x^\star}\log\frac{(x^\star)^2}{4x_0\varepsilon}$ we have*

$$x^\star - \varepsilon \leq x_t \leq x^\star.$$

*Proof.*

1. To prove the first part we simply need to apply Lemma B.7 $\left\lceil\log_2\frac{|x^\star - x_0|}{\varepsilon}\right\rceil$ times. Hence after

$$\frac{\log_2 e}{4\eta x^\star}\log\frac{|x^\star - x_0|}{\varepsilon} \leq \frac{3}{8\eta x^\star}\log\frac{|x^\star - x_0|}{\varepsilon}$$

iterations we are done.

2. We first need to find a lower-bound on time $t$ which ensures that $x_t \geq \frac{x^\star}{2}$. Note that while $x_t < \frac{x^\star}{2}$ we have $x_{t+1} \geq x_t(1 + 2\eta x^\star)^2$. Hence it is enough to choose a big enough $t$ such that

$$x_0(1 + 2\eta x^\star)^{2t} \geq \frac{x^\star}{2}$$

$$\Longleftrightarrow t \geq \frac{1}{2}\frac{\log\frac{x^\star}{2x_0}}{\log(1 + 2\eta x^\star)}.$$

We can upper-bound the term on the right by using $\log x \geq \frac{x-1}{x}$ as follows

$$\frac{1}{2}\frac{\log\frac{x^\star}{2x^\star}}{\log(1 + 2\eta x^\star)} \leq \frac{1}{2}\frac{1 + 2\eta x^\star}{2\eta x_0}\log\frac{x^\star}{2x_0}$$

$$\leq \frac{5}{16\eta x^\star}\log\frac{x^\star}{2x_0}$$

and so after $t \geq \frac{5}{16\eta x^{\star}} \log \frac{x^{\star}}{2x_0}$ we have $x_t \geq \frac{x^{\star}}{2}$.

Now we can apply the first part to finish the proof. The total sufficient number of iterations is then

$$\frac{5}{16\eta x^{\star}} \log \frac{x^{\star}}{2x_0} + \frac{3}{8\eta x^{\star}} \log \frac{x^{\star}}{2\varepsilon} \leq \frac{3}{8\eta x^{\star}} \log \frac{x^{\star}}{2x_0} + \frac{3}{8\eta x^{\star}} \log \frac{x^{\star}}{2\varepsilon}$$

$$= \frac{3}{8\eta x^{\star}} \log \frac{(x^{\star})^2}{4x_0\varepsilon}.$$

$\square$

We are now able to answer the question that we set out at the beginning of this section. That is, under what conditions on $x^{\star}, y^{\star}, \alpha$ and $\varepsilon$ does the inequality $T^x_{x^{\star}-\varepsilon} \leq T^y_{\alpha}$ hold? Let $\eta \leq \frac{1}{8x^{\star}}$ and suppose that $x^{\star} \geq 12y^{\star} > 0$. Lemmas B.6 and B.8 then tell us, that for any $\varepsilon > 0$ and any

$$t \geq \frac{12}{32\eta x^{\star}} \log \frac{(x^{\star})^2}{\alpha^2 \varepsilon}$$

the sequence $x_t$ has converged up to precision $\varepsilon$. Hence

$$T^x_{x^{\star}-\varepsilon} \leq \frac{12}{32\eta x^{\star}} \log \frac{(x^{\star})^2}{\alpha^2 \varepsilon} \tag{6}$$

On the other hand, we can now apply Lemma A.2 to see that for any

$$t \leq \frac{12}{32\eta x^{\star}} \log \frac{1}{\alpha^4} \leq \frac{1}{32\eta y^{\star}} \log \frac{1}{\alpha^4}$$

we have $y_t \leq \alpha$ and hence

$$T^y_{\alpha} \geq \frac{12}{32\eta x^{\star}} \log \frac{1}{\alpha^4} \tag{7}$$

We can now see from equations (6) and (7) that it is enough to set $\alpha \leq \frac{\sqrt{\varepsilon}}{x^{\star}}$ so that $T^x_{x^{\star}-\varepsilon} \leq T^y_{\alpha}$ is satisfied which answers our question.

## B.2 Dealing With Bounded Errors

In Section B.1 we analyzed one dimensional multiplicative update sequences and proved that it is possible to fit large enough signal while fitting a controlled amount of error. In this section we extend the setting considered in Section B.1 to handle bounded error sequences $(\mathbf{b}_t)_{t \geq 0}$ such that for any $t \geq 0$ we have $\|\mathbf{b}_t\|_{\infty} \leq B$ for some $B \in \mathbb{R}$. That is, we look at one-dimensional multiplicative sequences with positive target $x^{\star}$ with updates given by

$$x_{t+1} = x_t(1 - 4\eta(x_t - x^{\star} + b_t))^2. \tag{8}$$

Surely, if $B \geq x^{\star}$ one could always set $b_t = x^{\star}$ so that the sequence given with the above updates equation shrinks to 0 and convergence to $x^{\star}$ is not possible. Hence for a given $x^{\star}$ our lemmas below will require $B$ to be small enough, with a particular choice $B \leq \frac{1}{5}x^{\star}$. For a given $B$ one can only expect the sequence $(x_t)_{t \geq 0}$ to converge to $x^{\star}$ up to precision $B$. To see that, consider two extreme scenarios, one where for all $t \geq 0$ we have $b_t = B$ and another with $b_t = -B$. This gives rise the following two sequences with updates given by

$$x^-_{t+1} = x^-_t(1 - 4\eta(x^-_t - (x^{\star} - B)))^2,$$
$$x^+_{t+1} = x^+_t(1 - 4\eta(x^-_t - (x^{\star} + B)))^2. \tag{9}$$

We can think of sequences $(x^-_t)_{t \geq 0}$ and $(x^+_t)_{t \geq 0}$ as sequences with no errors and targets $x^{\star} - B$ and $x^{\star} + B$ respectively. We already understand the behavior of such sequences with the lemmas derived in Section B.1. The following lemma is the key result in this section. It tells us that the sequence $(x_t)_{t \geq 0}$ is sandwiched between sequences $(x^-_t)_{t \geq 0}$ and $(x^+_t)_{t \geq 0}$ for all iterations $t$. See Figure 6 for a graphical illustration.

Figure 6: A graphical illustration of Lemmas B.9 and B.10. For a given error bound $B$ we have sampled error sequence $(b_t)_{t \geq 0}$ from Uniform$[-B, B]$ distribution. Note that for $B = 0$ the above plots illustrate Lemma B.6.

**Lemma B.9** (Squeezing iterates with bounded errors). *Let $(b_t)_{t \geq 0}$ be a sequence of errors such that exists some $B > 0$ such that for all $t \geq 0$ we have $|b_t| \leq B$. Consider the sequences $(x_t^-)_{t \geq 0}$, $(x_t)_{t \geq 0}$ and $(x_t^+)_{t \geq 0}$ as defined in equations* (8) *and* (9) *with*

$$0 < x_0^- = x_0^+ = x_0 \leq x^\star + B$$

*If $\eta \leq \frac{1}{16(x^\star + B)}$ then for all $t \geq 0$*

$$0 \leq x_t^- \leq x_t \leq x_t^+ \leq x^\star + B.$$

*Proof.* We will prove the claim by induction. The claim holds trivially for $t = 0$. Then if $x_t^+ \geq x_t$, denoting $\Delta := x_t^+ - x_t \geq 0$ and $m_t := 1 - 4\eta(x_t - x^\star + b_t)$ we have

$$
\begin{aligned}
x_{t+1}^+ &= x_t^+ (1 - 4\eta(x_t^+ - x^\star - B))^2 \\
&= (x_t + \Delta)(1 - 4\eta(x_t - x^\star + b_t) - 4\eta(\Delta - B - b_t))^2 \\
&\geq (x_t + \Delta)(m_t - 4\eta\Delta)^2 \\
&= (x_t + \Delta)(m_t^2 - 8\eta\Delta m_t + 16\eta^2\Delta^2) \\
&\geq (x_t + \Delta)(m_t^2 - 8\eta\Delta m_t) \\
&= x_{t+1} + \Delta m_t^2 - x_t^+ 8\eta\Delta m_t \\
&= x_{t+1} + \Delta m_t (m_t - 8\eta x_t^+) \\
&\geq x_{t+1},
\end{aligned}
$$

where the last line is true since by lemma B.6 we have $0 < x_t^+ \leq x^\star + B$ and so using $\eta \leq \frac{1}{16(x^\star + B)}$ we get

$$
\begin{aligned}
m_t - 8\eta x_t^+ &\geq m_t - \frac{1}{2} \\
&= \frac{1}{2} - 4\eta(x_t - x^\star + b_t) \\
&\geq \frac{1}{2} - 4\eta(x^\star + B - x^\star + b_t) \\
&\geq \frac{1}{2} - 8\eta B \\
&\geq 0.
\end{aligned}
$$

Showing that $x_{t+1} \geq x_{t+1}^-$ follows a similar argument.

Finally, as we have already pointed out $x_t^+ \leq x^\star + B$ holds for all $t$ by the choice of $\eta$ and Lemma B.6. By induction and the choice of the step size we then also have for all $t \geq 0$

$$
\begin{aligned}
x_{t+1}^- = x_t^- (1 - 4\eta(x_t^- - x^\star + B))^2 \\
\geq x_t^- (1 - 8\eta B)^2 \\
\geq 0,
\end{aligned}
$$

which completes our proof. $\qquad\square$

Using the above lemma we can show analogous results for iterates with bounded errors to the ones shown in Lemmas B.6, B.7 and B.8.

We will first prove a counterpart to Lemma B.6, which is a crucial result in proving Proposition 1. As illustrated in Figure 6, monotonicity will hold while $|x_t - x^\star| > B$. On the other hand, once $x_t$ hits the $B$-tube around $x^\star$ it will always stay inside the tube. This is formalized in the next lemma.

**Lemma B.10** (Iterates with bounded errors monotonic behaviour)**.** *Consider the setting of Lemma B.9 with $B \leq \frac{1}{5}x^\star$, $\eta \leq \frac{5}{96x^\star}$ and $0 < x_0 \leq \frac{6}{5}x^\star$. Then the following holds:*

1. *If $|x_t - x^\star| > B$ then $|x_{t+1} - x^\star| < |x_t - x^\star|$.*

2. *If $|x_t - x^\star| \leq B$ then $|x_{t+1} - x^\star| \leq B$.*

*Proof.* First, note that our choice of step size, maximum error $B$ and maximum value for $x_0$ ensures that we can apply the second part of Lemma B.6 to the sequence $(x_t^-)_{t \geq 0}$ and the first part of Lemma B.6 to the sequence $(x_t^+)_{t \geq 0}$.

To prove the first part, note that if $0 < x_t < x^\star - B$ then $x_t < x_{t+1} \leq x_{t+1}^+ \leq x^\star + B$ and the result follows. On the other hand, if $x^\star + B < x_t \leq \frac{6}{5}x^\star$ then applying Lemma B.9 (with a slight abuse of notation, setting $x_0 := x_t$) we get $x^\star - B \leq x_{t+1}^- \leq x_{t+1} < x_t$ which finishes the proof of the first part.

The second part is immediate by Lemma B.9 applied again with a slight abuse of notation setting $x_0 := x_t$ and observing that by monotonicity Lemma B.6 the sequence $(x_t^-)_{t \geq 0}$ will monotonically decrease to $x^\star - B$ and the sequence $(x_t^+)_{t \geq 0}$ will monotonically increase to $x^\star + B$. $\qquad\square$

**Lemma B.11** (Iterates with bounded errors behaviour near convergence)**.** *Consider the setting of Lemma B.10. Then the following holds:*

1. *If $\frac{1}{2}(x^\star - B) \leq x_0 \leq x^\star - 5B$ then for any $t \geq \frac{5}{8\eta x^\star}$ we have*

$$
|x^\star - x_t| \leq \frac{1}{2}|x_0 - x^\star|.
$$

2. *If $x^\star + 4B < x_0 < \frac{6}{5}x^\star$ then for any $t \geq \frac{1}{4\eta x^\star}$ we have*

$$
|x^\star - x_t| \leq \frac{1}{2}|x_0 - x^\star|.
$$

*Proof.* Let the sequences $(x_t^+)_{t \geq 0}$ and $(x_t^-)_{t \geq 0}$ be given as in Lemma B.9. For the first part, we apply Lemma B.7 to the sequence $x_t^-$ twice, to get that for all

$$
t \geq \frac{5}{8\eta x^\star} \geq 2\frac{1}{4\eta(x^\star - B)}
$$

we have

$$
\begin{aligned}
0 \leq (x^\star - B) - x_t^- \\
\leq \frac{1}{4}|x_0 - (x^\star - B)| \\
\leq \frac{1}{4}|x_0 - x^\star| + \frac{1}{4}B.
\end{aligned}
$$

Then, if $x_t \le x^\star$ we have by Lemma B.9 and the above inequality

$$0 \le x^\star - x_t$$
$$\le x^\star - x_t^-$$
$$\le \frac{1}{4} |x_0 - x^\star| + \frac{5}{4} B$$
$$\le \frac{1}{2} |x_0 - x^\star| .$$

If $x_t \ge x^\star$ then by lemma B.9 we have

$$0 \le x_t - x^\star \le B \le \frac{1}{5} |x_0 - x^\star| ,$$

where the last inequality follows from $x_0 \le x^* - 5B$. This concludes the first part.

The second part can be shown similarly. We apply lemma B.7 to the sequence $x_t^+$ twice, to get that for all

$$t \ge 2 \frac{1}{8\eta x^\star} \ge 2 \frac{1}{8\eta(x^\star + B)}$$

we have

$$0 \le x_t^+ - (x^\star + B)$$
$$\le \frac{1}{4} |x_0 - (x^\star + B)|$$
$$\le \frac{1}{4} |x_0 - x^\star| + \frac{1}{4} B.$$

Then again, if $x_t \ge x^\star$ then

$$0 \le x_t - x^\star$$
$$\le x_t^+ - x^\star$$
$$\le \frac{1}{4} |x_0 - x^\star| + \frac{5}{4} B$$
$$\le \frac{1}{2} |x_0 - x^\star|$$

and if $x_t \le x^\star$ then by lemma B.9 we have

$$0 \le x^\star - x_t \le B \le \frac{1}{4} |x_0 - x^\star|$$

which finishes our proof. □

**Lemma B.12** (Iterates with bounded errors approach target exponentially fast)**.** *Consider the setting of Lemma B.10 and fix any $\varepsilon > 0$. Then the following holds:*

1. *If $B + \varepsilon < |x^\star - x_0| \le \frac{1}{5} x^\star$ then for any $t \ge \frac{15}{32\eta x^\star} \log \frac{|x^\star - x_0|}{\varepsilon}$ iterations we have $|x^\star - x_t| \le B + \varepsilon$.*

2. *If $0 < x_0 \le x^\star - B - \varepsilon$ then for any $t \ge \frac{15}{32\eta x^\star} \log \frac{(x^\star)^2}{x_0 \varepsilon}$ we have $x^\star - B - \varepsilon \le x_t \le x^\star + B$.*

*Proof.*

1. If $x_0 > x^\star + B$ then by Lemmas B.9 and B.10 we only need show that $(x_t^+)_{t \ge 0}$ hits $x^\star + B + \varepsilon$ within the desired number of iterations. By the first part of Lemma B.8 applied to the sequence $(x_t^+)_{t \ge 0}$ we see that $\frac{3}{8\eta(x^\star + B)} \log \frac{|x_0 - (x^\star + B)|}{\varepsilon} \le \frac{15}{32\eta x^\star} \log \frac{|x^\star - x_0|}{\varepsilon}$ iterations enough.

   Similarly, if $x_0 < x^\star - B$ by the first part of Lemma B.8 applied to the sequence $(x_t^-)_{t \ge 0}$ we see that $\frac{3}{8\eta(x^\star - B)} \log \frac{|x_0 - (x^\star - B)|}{\varepsilon} \le \frac{15}{32\eta x^\star} \log \frac{|x^\star - x_0|}{\varepsilon}$ iterations enough.

2. The upper-bound is immediate from lemma B.9. To get the lower-bound we simply apply the second part of lemma B.8 to the sequence $(x_t^-)_{t\geq 0}$ given in lemma B.9 to get that for any

$$ t \geq \frac{3}{8\eta \frac{4}{5} x^\star} \log \frac{(x^\star)^2}{x_0 \varepsilon} \geq \frac{3}{8\eta(x^\star - B)} \log \frac{(x^\star - B)^2}{x_0 \varepsilon} $$

we have $x^\star - B - \varepsilon \leq x_t^- \leq x_t$ which is what we wanted to show.

$\square$

## B.3 Dealing With Errors Proportional to Convergence Distance

In this section we derive lemmas helping to deal with errors proportional to convergence distance, that is, the error sequence $(\mathbf{p}_t)_{t\geq 0}$ given in equation (3) in Appendix A.1. Note that we cannot simply upper-bound $\|\mathbf{b}_t\|_\infty + \|\mathbf{p}_t\|_\infty$ by some large number independent of $t$ and treat both errors together as a bounded error sequence since $\|\mathbf{p}_0\|_\infty$ can be much larger than some of the coordinates of $\mathbf{w}^\star$. On the other hand, by Sections B.1 and B.2 we expect $\|\mathbf{s}_t - \mathbf{w}^\star\|_\infty$ to decay exponentially fast and hence the error $\|\mathbf{p}_t\|_\infty$ should also decay exponentially fast.

Let $m$ and $T_0, \ldots, T_{m-1}$ be some integers and suppose that we run gradient descent for $\sum_{i=0}^{m-1} T_i$ iterations. Suppose that for each time interval $\sum_{i=0}^{j-1} T_i \leq t \leq \sum_{i=0}^{j} T_i$ we can upper-bound $\|\mathbf{b}_t\|_\infty + \|\mathbf{p}_t\|_\infty$ by $2^{-j}B$ for some $B \in \mathbb{R}$. The following lemma then shows how to control errors of such type and it is, in fact, the reason why in the main theorems a logarithmic term appears in the upper-bounds for the RIP parameter $\delta$. We once again restrict ourselves to one-dimensional sequences.

**Lemma B.13** (Halving errors over doubling time intervals). *Let $T > 0$ be some fixed positive real number, $T_i := 2^i T$ and $\bar{T}_i := \sum_{j=0}^{i} T_j$. Further, suppose $(p_t)_{t\geq 0}$ is a sequence of real numbers and let $B \in \mathbb{R}$. Suppose that for every integer $i \geq 0$ and for any $\bar{T}_{i-1} \leq t < \bar{T}_i$ we have $|p_t| \leq 2^{-i}B$. Then, for any integer $i \geq 0$ and $\eta \leq \frac{1}{4B}$*

$$ \prod_{i=0}^{\bar{T}_i - 1} (1 + 4\eta p_t)^2 \leq (1 + 4\eta 2^{-i}B)^{2(i+1)T_i}. $$

*Proof.* Note that for $x, y \geq 0$ we have $(1+x+y) \leq (1+x)(1+y)$ and in particular, for any integers $i \geq j \geq 0$

$$ 1 + 4\eta 2^{-j}B \leq (1 + 4\eta 2^{-j-1}B)^2 \leq \cdots \leq (1 + 4\eta 2^{-i}B)^{2^{i-j}}. $$

It follows that

$$ \prod_{t=0}^{\bar{T}_i - 1} (1 + 4\eta p_t)^2 \leq \prod_{j=0}^{i} (1 + 4\eta 2^{-j}B)^{2T_j} $$

$$ \leq \prod_{j=0}^{i} (1 + 4\eta 2^{-i}B)^{2^{i-j} 2 T_j} $$

$$ = (1 + 4\eta 2^{-i}B)^{2(i+1)T_i}. $$

$\square$

Sometimes $\|\mathbf{p}_t\|_\infty$ can be much larger than some coordinates of the true parameter vector $\mathbf{w}^\star$. For example, if $w_{\max}^\star \gg w_{\min}^\star$ then $\|\mathbf{p}_0\|_\infty$ can be much larger than $w_{\min}^\star$. In Section B.2 we have shown how to deal with bounded errors that are much smaller than target. We now show how to deal with errors much larger than the target.

**Lemma B.14** (Handling large errors). *Let $(b_t)_{t\geq 0}$ be a sequence of errors such that for some $B \in \mathbb{R}$ and all $t \geq 0$ we have $|b_t| \leq B$. Consider a sequence defined as*

$$ x^\star + 2B \leq x_0 \leq x^\star + 4B, $$

$$ x_{t+1} = x_t(1 - 4\eta(x_t - x^\star + b_t))^2. $$

*Then, for $\eta \leq \frac{1}{20B}$ and any $t \geq \frac{1}{10\eta B}$ we have*

$$0 \leq x_t \leq x^\star + 2B.$$

*Proof.* Note that if $x_t \geq x^\star + 2B$ then

$$x_{t+1} = x_t(1 - 4\eta(x_t - x^\star + b_t))^2$$
$$\leq x_t(1 - 4\eta B)^2.$$

Hence to find $t$ such that $x_t \leq x^\star + 2B$ it is enough to satisfy the following inequality

$$(x^\star + 4B)(1 - 4\eta B)^{2t} \leq x^\star + 2B$$
$$\iff t \geq \frac{1}{2}\frac{1}{\log(1 - 4\eta B)}\log\frac{x^\star + 2B}{x^\star + 4B}$$

Since for any $x \in (0,1)$ we have $\log(1-x) \leq -x$ hence $\log(1 - 4\eta B) \leq -4\eta B$. Also, since $\frac{x^\star+2B}{x^\star+4B} \geq \frac{1}{2}$ we have $\log\frac{x^\star+2B}{x^\star+4B} \geq \log\frac{1}{2} \geq -\frac{7}{10}$. Hence

$$\frac{1}{2}\frac{1}{\log(1 - 4\eta B)}\log\frac{x^\star + 2B}{x^\star + 4B} \leq \frac{1}{2} \cdot \frac{1}{-4\eta B} \cdot \frac{-7}{10}.$$

Setting $t \geq \frac{1}{10\eta B}$ is hence enough. To ensure non-negativity of the iterates, note that

$$|4\eta(x_t - x^\star + b_t)| \leq 20\eta B$$

and hence setting $\eta \leq \frac{1}{20B}$ is enough. $\qquad \square$

The final challenge caused by the error sequence $(\mathbf{p}_t)_{t \geq 0}$ is that some of the signal components $\mathbf{1}_S \odot \mathbf{w}_t$ can actually shrink initially instead of approaching the target. Hence for all $t \geq 0$ we need to control the maximum shrinkage by bounding the following term from below

$$\alpha^2 \prod_{i=0}^{t-1}(1 - 4\eta(\|\mathbf{b}_t\|_\infty + \|\mathbf{p}_t\|_\infty))^2. \tag{10}$$

Recall that we are handling maximum growth of the error sequence $(\mathbf{e}_t)_{t \geq 0}$ by Lemma A.1 which requires upper-bounding the term

$$\alpha^2 \prod_{i=0}^{t-1}(1 + 4\eta(\|\mathbf{b}_t\|_\infty + \|\mathbf{p}_t\|_\infty))^2. \tag{11}$$

If the term in equation (11) is not too large, then we can prove that the term in equation (10) cannot be too small. This idea is exploited in the following lemma.

**Lemma B.15** (Handling signal shrinkage). *Consider a sequence*

$$x_0 = \alpha^2,$$
$$x_{t+1} = x_t(1 - 4\eta(x^\star + b_t + p_t))^2$$

*where $x^\star > 0$ and exists some $B > 0$ such that for all $t \geq 0$ we have $|b_t| + |p_t| \leq B$. If $\eta \leq \frac{1}{8B}$ and*

$$\prod_{i=0}^{t-1}(1 + 8\eta(|b_t| + |p_t|))^2 \leq \frac{1}{\alpha}$$

*then*

$$\prod_{i=0}^{t-1}(1 - 4\eta(|b_t| + |p_t|))^2 \geq \alpha.$$

*Proof.* By the choice of step size $\eta$ we always have $0 \leq 4\eta(|b_t| + |p_t|) \leq \frac{1}{2}$. Since for $x \in [0, \frac{1}{2}]$ we have $(1 + 2x)(1 - x) = 1 + x - 2x^2 \geq 1$ it follows that

$$\prod_{i=0}^{t-1}(1 + 8\eta(|b_t| + |p_t|))^2 \prod_{i=0}^{t-1}(1 - 4\eta(|b_t| + |p_t|))^2 \geq 1$$

and we are done. $\qquad \square$

## B.4 Dealing With Negative Targets

So far we have only dealt with sequences converging to some positive target, i.e., the parametrization $\mathbf{w}_t = \mathbf{u}_t \odot \mathbf{u}_t$. In this section we show that handling parametrization $\mathbf{w}_t = \mathbf{u}_t \odot \mathbf{u}_t - \mathbf{v}_t \odot \mathbf{v}_t$ can be done by noting that for any coordinate $i$, at least one of $u_{t,i}$ or $v_{t,i}$ has to be close to its initialization value. Intuitively, this observation will allow us to treat parametrization $\mathbf{w}_t = \mathbf{u}_t \odot \mathbf{u}_t - \mathbf{v}_t \odot \mathbf{v}_t$ as if it was $\mathbf{w}_t \approx \mathbf{u}_t \odot \mathbf{u}_t$ and all coordinates of the target $\mathbf{w}^\star$ are replaced by its absolute values.

Consider two sequences given by

$$0 < x_0^+ = \alpha^2 \le x_+^\star, \quad x_{t+1}^+ = x_t^+ (1 - 4\eta(x_t^+ - x_+^\star + b_t))^2$$
$$0 < x_0^- = \alpha^2 \le -x_-^\star, \quad x_{t+1}^- = x_t^- (1 + 4\eta(-x_t^- - x_-^\star + b_t))^2$$

where $(b_t)_{t\ge0}$ is some sequence of errors and the targets satisfy $x_+^\star > 0$ and $x_-^\star < 0$. We already know how to deal with the sequence $(x_t^+)_{t\ge0}$. Note that we can rewrite the updates for the sequence $(x_t^-)_{t\ge0}$ as follows

$$x_{t+1}^- = x_t^- (1 - 4\eta(x_t^- - \left|x_-^\star\right| - b_t))^2.$$

and we know how to deal with sequences of this form. In particular, $(x_t^-)_{t\ge0}$ will converge to $\left|x_-^\star\right|$ with error at most $B$ equal to some bound on maximum error and hence the sequence $(-x_t^-)_{t\ge0}$ will converge to a $B$-tube around $x^\star$. Hence, our theory developed for sequences with positive targets directly apply for sequences with negative targets of the form given above.

The following lemma is the key result allowing to treat $\mathbf{w}_t = \mathbf{u}_t \odot \mathbf{u}_t - \mathbf{v}_t \odot \mathbf{v}_t$ almost as if it was $\mathbf{w}_t \approx \mathbf{u}_t \odot \mathbf{u}_t$ as discussed at the beginning of this section.

**Lemma B.16** (Handling positive and negative sequences simultaneously). *Let $x_t = x_t^+ - x_t^-$ and $x^\star \in \mathbb{R}$ be the target such that $|x^\star| > 0$. Suppose the sequences $(x_t^+)_{t\ge0}$ and $(x_t^-)_{t\ge0}$ evolve as follows*

$$0 < x_0^+ = \alpha^2 \le \frac{1}{4}|x^\star|, \quad x_{t+1}^+ = x_t^+ (1 - 4\eta(x_t - x^\star + b_t))^2$$

$$0 < x_0^- = \alpha^2 \le \frac{1}{4}|x^\star|, \quad x_{t+1}^- = x_t^- (1 + 4\eta(x_t - x^\star + b_t))^2.$$

*and that there exists $B > 0$ such that $|b_t| \le B$ and $\eta \le \frac{1}{12(x^\star + B)}$. Then the following holds:*

1. *For any $t \ge 0$ we have $0 \le x_t^+ \wedge x_t^- \le \alpha^2$.*

2. *For any $t \ge 0$ we have*

   - *If $x^\star > 0$ then $x_t^- \le \alpha^2 \prod_{i=0}^{t-1}(1 + 4\eta|b_t|)$.*
   - *If $x^\star < 0$ then $x_t^+ \le \alpha^2 \prod_{i=0}^{t-1}(1 + 4\eta|b_t|)$.*

*Proof.* The choice of our step size ensures that $|4\eta(x_t - x^\star + b_t)| \le \frac{1}{2}$. For any $0 \le a \le \frac{1}{2}$ we have $0 \le (1-a)(1+a) = 1 - a^2 \le 1$. In particular, this yields for any $t \ge 0$

$$x_t^+ x_t^- = \alpha^4 \prod_{i=0}^{t-1}(1 - 4\eta(x_t - x^\star + b_t))^2 (1 + 4\eta(x_t - x^\star + b_t))^2 \le \alpha^4$$

which concludes the first part.

To prove the second part assume $x^\star > 0$ and fix any $t \ge 0$. Let $0 \le s \le t$ be the largest $s$ such that $x_s^+ > x^\star$. If no such $s$ exists we are done immediately. If $s = t$ then by the first part we have $x_t^- \le \alpha^2$ and we are done.

If $s < t$ then we have by the first part and by the assumption $\alpha^2 \le \frac{1}{4}|x^\star|$, $x_s^- \le \frac{\alpha^4}{x_s^+} \le \frac{1}{4}\alpha^2$. Further, by the choice of step size $\eta$ we have $x_s^+ \le 4x^\star$. It then follows that

$$(1 + 4\eta(x_s - x^\star + b_t))^2 \le 4$$

and hence

$$x_t^- = x_s^- \prod_{i=s}^{t-1} (1 + 4\eta(x_i^+ - x_i^- - x^\star + b_i))^2$$

$$\leq \frac{1}{4}\alpha^2(1 + 4\eta(x_s - x^\star + b_t))^2 \prod_{i=s+1}^{t-1} (1 + 4\eta(x_i^+ - x_i^- - x^\star + b_i))^2$$

$$\leq \alpha^2 \prod_{i=s+1}^{t-1} (1 + 4\eta|b_t|))^2.$$

This completes our proof for the case $x^\star > 0$. For $x^\star < 0$ we are done by symmetry. $\qquad\square$

### B.5 Proof of Proposition 1

In this section we will prove Proposition 1. We remind our readers, that the goal of this proposition is showing that the error sequence $(\mathbf{p}_t)_{t\geq 0}$ can be essentially ignored if the RIP constant $\delta$ is small enough.

Recall that the error arising due to the bounded error sequence $(\mathbf{b}_t)_{t\geq 0}$ is irreducible as discussed in Section B.2. More formally, we will show that if for some $0 \leq \zeta \leq w^\star_{\max}$ we have $\|\mathbf{b}_t\|_\infty \lesssim \zeta$ and if $\|\mathbf{p}_t\|_\infty \lesssim \frac{1}{\log_2 \frac{w^\star_{\max}}{\zeta}} \|s_t - \mathbf{w}^\star\|_\infty$ then after $t = O\left(\frac{1}{\eta\zeta}\log\frac{1}{\alpha}\right)$ iterations we have $\|\mathbf{s}_t - \mathbf{w}^\star\|_\infty \leq \zeta$. In particular, up to absolute multiplicative constants we perform as good as if the error sequence $(\mathbf{p}_t)_{t\geq 0}$ was equal to 0.

The proof idea is simple, but the details can complicated. We will first prove a counterpart to Proposition 1 which will correspond to parametrization $\mathbf{w}_t = \mathbf{u}_t \odot \mathbf{u}_t$, that is, we will only try to fit the positive coordinates of $\mathbf{w}^\star$. We will later use Lemma B.16 to extend our result to the general case. We now list the key ideas appearing in the proof below.

1. Initially we have $\|\mathbf{w}_0 - \mathbf{w}^\star\|_\infty \leq w^\star_{\max}$. We will prove our claim by induction, reducing the above distance by half during each induction hypothesis. We will hence need to apply $m := \left\lceil \log_2 \frac{w^\star_{\max}}{\zeta}\right\rceil$ induction steps which we will enumerated from 0 to $m-1$.

2. At the beginning of the $i^{th}$ induction step we will have $\|\mathbf{w}_t - \mathbf{w}^\star\|_\infty \leq 2^{-i}w^\star_{\max}$. Choosing small enough absolute constants for upper-bounds on error sequences $(\mathbf{b}_t)_{t\geq 0}$ and $(\mathbf{p}_t)_{t\geq 0}$ we can show that
$$\|\mathbf{b}_t\|_\infty + \|\mathbf{p}_t\|_\infty \leq \frac{1}{40}2^{-i}w^\star_{\max} =: B_i.$$
In particular, during the $i^{th}$ induction step we treat both types of errors simultaneously as a bounded error sequence with bound $B_i$. Since at each induction step $\|\mathbf{w}_t - \mathbf{w}^\star\|_\infty$ decreases by half, the error bound $B_i$ also halves. This puts us in position to apply Lemma B.13 which plays a key role in the proof below.

3. One technical difficulty is that in Section B.2 all lemmas require that iterates never exceed the target by more than a factor $\frac{6}{5}$. We cannot ensure that since initially our errors can be much larger than some of the true parameter $\mathbf{w}^\star$ coordinates. We instead use Lemma B.14 to show that for any coordinate $j$ we have $w_{t,j} \leq w^\star_j + 4B_i$ during $i^{th}$ induction step. Then for any $j$ such that $w^\star_j \geq 20B_i$ we can apply the results from Section B.2. On the other hand, if $w^\star_j \leq 20B_i = \frac{1}{2}2^{-i}w^\star_{\max}$ then we already have $\left|w_{t,j} - w^\star_j\right| \leq 2^{-i-1}w^\star_{\max}$ and the above bound does not change during the $i^{th}$ induction step.

4. During the $i^{th}$ induction step, if $\left|w_{t,j} - w^\star_j\right| > 2^{-i-1}w^\star_{\max}$ then $w^\star_j \geq 20B_i$ and we can apply Lemma B.10 which says that all such coordinates will monotonically approach $B$-tube around $w^\star_j$. Lemma B.12 then tells us how many iterations need to be taken for our iterates to get close enough to this $B$-tube so that $\left|w_{t,j} - w^\star_j\right| \leq 2^{-i-1}w^\star_{\max}$.

5. Finally, we control the total accumulation of errors $\prod_{i=0}^{t-1}(1 + 4\eta(\|\mathbf{b}_i\|_\infty + \|\mathbf{p}_i\|_\infty))^2$ using Lemma B.13 and ensure that for any $w^\star_j \geq 0$ the iterates never get below $\alpha^3$ by applying Lemma B.15.

**Lemma B.17** (Dealing with errors proportional to convergence distance). *Fix any $0 < \zeta \le w^\star_{\max}$ and let $\gamma = \frac{C_\gamma}{\lceil \log_2 \frac{w^\star_{\max}}{\zeta} \rceil}$ where $C_\gamma$ is some small enough absolute constant. Let $\mathbf{w}^\star \in \mathbb{R}^k$ be a target vector which is now allowed to have negative components. Denote by $\mathbf{w}^\star_+$ the positive part of $\mathbf{w}^\star$, that is, $(w^\star_+)_i = \mathbb{1}_{\{w^\star_i \ge 0\}} w^\star_i$. Let $(\mathbf{b}_t)_{t \ge 0}$ and $(\mathbf{p}_t)_{t \ge 0}$ sequences of errors such that for all $t \ge 0$ we have $\|\mathbf{b}_t\|_\infty \le C_b \zeta$ for some small enough absolute constant $C_b$ and $\|\mathbf{p}_t\|_\infty \le \gamma \|\mathbf{w}_t - \mathbf{w}^\star_+\|_\infty$. Let the updates be given by*

$$w_{0,j} = \alpha^2, \quad w_{t+1,j} = w_{t,j}(1 - 4\eta(w_{t,j} - w^\star_j + b_{t,j} + p_{t,j}))^2.$$

*If the step size satisfies $\eta \le \frac{5}{96 w^\star_{\max}}$ and the initialization satisfies $\alpha \le \frac{\zeta}{3(w^\star_{\max})^2} \wedge \sqrt{w^\star_{\min}} \wedge 1$ then for $t = O\left(\frac{1}{\eta\zeta} \log \frac{1}{\alpha}\right)$ we have*

$$\|\mathbf{w}_t - \mathbf{w}^\star_+\|_\infty \le \zeta$$

$$\alpha^2 \prod_{i=0}^{t-1}(1 + 4\eta(\|\mathbf{b}_t\|_\infty + \|\mathbf{p}_t\|_\infty))^2 \le \alpha.$$

*Proof.* Let $T := \frac{1}{\eta w^\star_{\max}} \log \frac{1}{\alpha^4}$ and for any integer $i \ge -1$ let $T_i := 2^i T$ and $\bar{T}_i := \sum_{j=0}^i T_j$. We also let $\bar{T}_{-1} = 0$. Let $B_i := \frac{1}{40} 2^{-i} w^\star_{\max}$. Let $m = \left\lceil \log_2 \frac{w^\star_{\max}}{\zeta} \right\rceil$ so that $\gamma = \frac{C_\gamma}{m}$. We will prove our claim by induction on $i = 0, 1, \dots, m-1$.

*Induction hypothesis for $i \in \{0, \dots, m\}$*

1. For any $j < i$ and $\bar{T}_{j-1} \le t < \bar{T}_j$ we have $\|\mathbf{w}_t - \mathbf{w}^\star_+\|_\infty \le 2^{-j} w^\star_{\max}$. In particular, this induction hypothesis says that we halve the convergence distance during each induction step.

2. We have $\left\|\mathbf{w}_{\bar{T}_{i-1}} - \mathbf{w}^\star_+\right\|_\infty \le 2^{-i} w^\star_{\max}$. This hypothesis controls the convergence distance at the beginning of the $i^{th}$ induction step.

3. For any $j$ we have $\alpha^3 \le w_{\bar{T}_{i-1},j} \le w^\star_j + 4B_i$.

*Base case*
For $i = 0$ all conditions hold since for all $j$ we have $0 \le \alpha^2 = w_{0,j} < w^\star_j$.

*Induction step*
Assume that the induction hypothesis holds for some $0 \le i < m$. We will show that it holds for $i + 1$.

1. We want to show that for all $t \in \{0, \dots, T_i - 1\}$ $\left\|\mathbf{w}_{\bar{T}_{i-1}+t} - \mathbf{w}^\star_+\right\|_\infty$ remains upper-bounded by $2^{-i} w^\star_{\max}$.

   Note that $2^{-i} w^\star_{\max} \ge 2^{-m} w^\star_{\max} \ge \frac{1}{2}\zeta$ and hence requiring $C_\gamma + 2C_b \le \frac{1}{40}$ we have

$$\left\|\mathbf{b}_{\bar{T}_{i-1}}\right\|_\infty + \left\|\mathbf{p}_{\bar{T}_{i-1}}\right\|_\infty \le C_b \zeta + \gamma 2^{-i} w^\star_{\max}$$
$$\le (C_\gamma + 2C_b) 2^{-i} w^\star_{\max}$$
$$\le \frac{1}{40} 2^{-i} w^\star_{\max}$$
$$= B_i.$$

   For any $j$ such that $w^\star_j \ge 20 B_i$ the third induction hypothesis $w_{\bar{T}_{i-1},j} \le w^\star_j + 4B_i$ ensures that $w_{\bar{T}_{i-1},j} \le \frac{6}{5} w^\star_j$. Hence, it satisfies the pre-conditions of Lemma B.10 and as long as

$$\left\|\mathbf{w}_{\bar{T}_{i-1}+t} - \mathbf{w}^\star_+\right\|_\infty \le 2^{-i} w^\star_{\max}$$

any such $j$ will monotonically approach the $\frac{1}{40}B_i$-tube around $w_j^\star$ maintaining $\left|w_t - w_j^\star\right| \leq 2^{-i}w_{\max}^\star$.

On the other hand, for any $j$ such that $w_j^\star \leq 20B_i$ $w_{t,j}$ will stay in $(0, w_j^\star + 4B_i]$ maintaining $\left|w_t - w_j^\star\right| \leq 20B_i \leq 2^{-i}w_{\max}^\star$ as required.

By induction on $t$, we then have for any $t \geq 0$

$$\left\|\mathbf{w}_{\bar{T}_{i-1}+t} - \mathbf{w}_+^\star\right\|_\infty \leq 2^{-i}w_{\max}^\star$$

which is what we wanted to show.

2. To prove the second part of the induction hypothesis, we need to show that after $T_i$ iterations the maximum convergence distance $\left\|\mathbf{w}_{\bar{T}_i} - \mathbf{w}_+^\star\right\|_\infty$ decreases at least by half.

   Take any $j$ such that $w_j^\star \geq 0$ and $\left|w_{\bar{T}_{i-1},j}^\star - w_j^\star\right| \leq 2^{-i-1}w_{\max}^\star = 20B_i$. Then by a similar argument used in to prove the first induction hypothesis for any $t \geq 0$ we have $\left|w_{\bar{T}_{i-1}+t,j}^\star - w_j^\star\right| \leq 2^{-i-1}w_{\max}^\star$ and hence such coordinates can be ignored.

   Now take any $j$ such that $w_j^\star \geq 0$ and $\left|w_{\bar{T}_{i-1},j}^\star - w_j^\star\right| > 2^{-i-1}w_{\max}^\star$. Then, since $20B_i = 2^{-i-1}w_{\max}^\star$ and since by the third induction hypothesis $w_{\bar{T}_{i-1},j} \leq w_j^\star + 4B_i$ it follows that $0 \leq w_{\bar{T}_{i-1},j} < w_j^\star - 20B_i$. Applying the second part of Lemma B.12 with $\varepsilon = 19B_i$ and noting that

$$19B_i = \frac{19}{40}2^{-i}w_{\max}^\star \geq \frac{19}{40}2^{-m+1}w_{\max}^\star \geq \frac{19}{40}\zeta \geq \frac{1}{3}\zeta$$

   we have for any

$$t \geq T_i$$
$$\geq 2^i\frac{1}{\eta w_{\max}^\star}\log\frac{3(w_{\max}^\star)^2}{\alpha^3\zeta}$$
$$\geq \frac{15}{32\eta w_j^\star}\log\frac{(w_j^\star)^2}{w_{\bar{T}_{i-1},j}\cdot 19B_i}$$

   iterations the following holds

$$\left|w_{\bar{T}_{i-1}+t,j} - w_j^\star\right| \leq 20B_i \leq 2^{-i-1}w_{\max}^\star$$

   which completes our proof.

3. The upper bound follows immediately from Lemma B.14 which tells that after

$$t \geq T_i \geq 2^i\frac{4}{\eta w_{\max}^\star} = \frac{1}{10\eta B_i}.$$

   iterations for any $j$ we have $w_{\bar{T}_{i-1}+t,j} \leq w_j^\star + 2B_i = w_j^\star + 4B_{i+1}$.

To prove the lower-bound, first note that

$$\prod_{i=0}^{\bar{T}_i-1} (1 + 8\eta(\|\mathbf{b}_i\|_\infty + \|\mathbf{p}_i\|_\infty))^2$$

$$\leq \prod_{i=0}^{\bar{T}_i-1} (1 + 8\eta C_b \zeta)^2 (1 + 4\eta \|\mathbf{p}_i\|_\infty))^4 \tag{12}$$

$$\leq (1 + 8\eta C_b \zeta)^{4T_i} \left(1 + 4\eta \cdot \frac{C_\gamma}{m} 2^{-i} w_{\max}^\star\right)^{4(i+1)T_i} \tag{13}$$

$$\leq (1 + 8\eta C_b \zeta)^{4T_{m-1}} \left(1 + 4\eta \cdot \frac{C_\gamma}{m} 2^{-m+1} w_{\max}^\star\right)^{4mT_{m-1}} \tag{14}$$

$$\leq \left(1 + 4\eta \cdot \frac{1}{m} 2C_b \zeta\right)^{4mT_{m-1}} \left(1 + 4\eta \cdot \frac{C_\gamma}{m} 2^{-m+1} w_{\max}^\star\right)^{4mT_{m-1}} \tag{15}$$

$$\leq \left(1 + 4\eta \cdot \frac{C_\gamma}{m} 2^{-m+1} w_{\max}^\star\right)^{8mT_{m-1}} \tag{16}$$

$$\leq \frac{1}{\alpha} \tag{17}$$

where line 12 follows by noting that for any $x, y \geq 0$ we have $(1 + x + y) \leq (1 + x)(1 + y)$. Line 13 follows by applying Lemma B.13 and noting that $\bar{T}_i \leq 2T_i$. Line 14 follows by noting that $i \leq m - 1$. Line 15 follows by applying $(1 + mx) \leq (1 + x)^m$ for $x \geq 0$ and $m \geq 1$. Line 16 follows by noting that $\zeta \leq 2^{-m+1} w_{\max}^\star$ and assuming that $2C_b \leq C_\gamma$. Line 17 follows by applying Lemma A.2 which in particular says that

$$\left(1 + 4\eta \cdot \frac{C_\gamma}{m} 2^{-m+1} w_{\max}^\star\right)^{2t} \leq \frac{1}{\alpha}$$

for any $t \leq \frac{m2^{m-1}}{32\eta w_{\max}^\star C_\gamma} \log \frac{1}{\alpha^4}$. Setting $C_\gamma = \frac{1}{128}$ yields the desired result.

The lower-bound is then proved immediately by Lemma B.15.

By above, the induction hypothesis holds for $i = m$. We can still repeat the argument for the first step of induction hypothesis to show that for any $t \geq \bar{T}_{m-1}$

$$\left\| \mathbf{w}_t - \mathbf{w}_+^\star \right\|_\infty \leq 2^{-m} w_{\max}^\star \leq \zeta.$$

Also, the proof for the third induction hypothesis with $i = m$ shows that for any $t \leq \bar{T}_{m-1}$ we have

$$\alpha^2 \prod_{i=0}^{t-1} (1 + 4\eta(\|\mathbf{b}_t\|_\infty + \|\mathbf{p}_t\|_\infty))^2 \leq \alpha.$$

To simplify the presentation, note that $\frac{w_{\max}^\star}{\zeta} \leq 2^m < \frac{2w_{\max}^\star}{\zeta}$ and hence we will write

$$\bar{T}_{m-1} = (2^m - 1) \frac{1}{\eta w_{\max}^\star} \log \frac{1}{\alpha^4} = O\left(\frac{1}{\eta\zeta} \log \frac{1}{\alpha}\right).$$

Finally, regarding the absolute constants we have required in our proofs above that $C_\gamma + 2C_b \leq \frac{1}{40}$, $C_b \leq \frac{1}{2}C_\gamma$ and $C_\gamma \leq \frac{1}{128}$. Hence, for example, absolute constants $C_b = \frac{1}{256}$ and $C_\gamma = \frac{1}{128}$ satisfy the requirements of this lemma. $\qquad \square$

Extending the above lemma to the general setting considered in Proposition 1 can now be done by a simple application of Lemma B.16 as follows.

*Proof of Proposition 1.* Lemma B.16 allows us to reduce this proof to lemma B.17 directly. In particular, using notation from Lemma B.17 and using Lemma B.16 we maintain that for all $t \leq \bar{T}_{m-1}$

$$w_j^\star > 0 \implies 0 \leq w_t^- \leq \alpha$$
$$w_j^\star < 0 \implies 0 \leq w_t^+ \leq \alpha.$$

Consequently, for $w_j^\star > 0$ we can ignore sequence $(w_{t,j}^-)_{t \geq 0}$ by treating it as a part of bounded error $b_t$. The same holds for sequence $(w_{t,j}^+)_{t \geq 0}$ when $w_j^\star < 0$. Then, for $w_j^\star > 0$ the sequence $(w_{t,j}^+)$ evolves as follows

$$w_{t+1,j}^+ = w_{t,j}^+(1 - 4\eta(w_{t,j}^+ - w_j^\star + (b_{t,j} - w_{t,j}^-) + p_{t,j}))^2$$

which falls directly into the setting of lemma B.17. Similarly, if $w_j^\star < 0$ then

$$w_{t+1,j}^- = w_{t,j}^-(1 + 4\eta(-w_{t,j}^- - w_j^\star + (b_{t,j} + w_{t,j}^+) + p_{t,j}))^2$$
$$= w_{t,j}^-(1 - 4\eta(w_{t,j}^- - |w_j^\star| + (-b_{t,j} - w_{t,j}^+) - p_{t,j}))^2$$

and hence this sequence also falls into the setting of lemma B.17.

Finally, $\|\mathbf{e}_t\|_\infty \leq \alpha$ follows by Lemma A.1 and we are done. $\qquad\square$

### B.6    Proof of Proposition 2

We split the proof of Proposition 2 in two phases. First, using Lemma B.18 we show that $\|\mathbf{s}_t - \mathbf{w}^\star\|_\infty$ converges to 0 with error $\|\mathbf{b}_t \odot \mathbf{1}_S\|_\infty$ up to some absolute multiplicative constant. From this point onward, we can apply Lemma B.12 to handle convergence to each individual sequence $i$ on the true support $S$ up to the error $\|\mathbf{b}_t \odot \mathbf{1}_i\|_\infty \vee \sqrt{k}\delta \|\mathbf{b}_t \odot \mathbf{1}_S\|_\infty$. This is exactly what allows us to approach an oracle-like performance with the $\ell_2$ parameter estimation error depending on $\log k$ instead of $\log d$ in the case of sub-Gaussian noise.

**Lemma B.18.** *Consider the setting of updates given in equations* (3) *and* (4)*. Fix any $\varepsilon > 0$ and suppose that the error sequences $(\mathbf{b}_t)_{t \geq 0}$ and $(\mathbf{p}_t)_{t \geq 0}$ satisfy the following for any $t \geq 0$:*

$$\|\mathbf{b}_t \odot \mathbf{1}_S\|_\infty \leq B,$$

$$\|\mathbf{p}_t\|_\infty \leq \frac{1}{20}\|\mathbf{s}_t - \mathbf{w}^\star\|_\infty.$$

*Suppose that*

$$20B < \|\mathbf{s}_0 - \mathbf{w}^\star\|_\infty \leq \frac{1}{5}w_{\min}^\star.$$

*Then for $\eta \leq \frac{5}{96 w_{\max}^\star}$ and any $t \geq \frac{5}{8\eta w_{\min}^\star}$ we have*

$$\|\mathbf{s}_t - \mathbf{w}^\star\|_\infty \leq \frac{1}{2}\|\mathbf{s}_0 - \mathbf{w}^\star\|_\infty.$$

*Proof.* Note that $\|\mathbf{b}_0\|_\infty + \|\mathbf{p}_0\|_\infty \leq \frac{1}{10}\|\mathbf{s}_0 - \mathbf{w}^\star\|_\infty$. By Lemma B.10 for any $t \geq 0$ we have $\|\mathbf{b}_t\|_\infty + \|\mathbf{p}_t\|_\infty \leq \frac{1}{10}\|\mathbf{s}_0 - \mathbf{w}^\star\|_\infty$. Hence, for any $i$ such that $|s_{0,i} - w_i^\star| \leq \frac{1}{2}\|\mathbf{s}_0 - \mathbf{w}^\star\|_\infty$ Lemma B.10 guarantees that for any $t \geq 0$ we have $|s_{t,i} - w_i^\star| \leq \frac{1}{2}\|\mathbf{s}_0 - \mathbf{w}^\star\|_\infty$ On the other hand, for any $i$ such that $|s_{0,i} - w_i^\star| > \frac{1}{2}\|\mathbf{s}_0 - \mathbf{w}^\star\|_\infty$ by Lemma B.11 we have $|s_{t,i} - w_i^\star| \leq \frac{1}{2}\|\mathbf{s}_0 - \mathbf{w}^\star\|_\infty$ for any $t \geq \frac{5}{8\eta w_{\min}^\star}$ which is what we wanted to prove. $\qquad\square$

*Proof of Proposition 2.*

Let $B := \max_{j \in S} B_j$. To see that $\|\mathbf{s}_t - \mathbf{w}^\star\|_\infty$ never exceeds $\frac{1}{5}w_{\min}^\star$ we use the $B$-tube argument developed in Section B.2 and formalized in Lemma B.10.

We begin by applying the Lemma B.18 for $\log_2 \frac{w_{\min}^\star}{5(B \vee \varepsilon)}$ times. Now we have $\|\mathbf{s}_t - \mathbf{w}^\star\|_\infty < 20(B \vee \varepsilon)$ and so $\|\mathbf{p}_t\|_\infty < \delta\sqrt{k} \cdot 20(B \vee \varepsilon)$ Hence, for any $i \in S$ we have

$$\|\mathbf{b}_t \odot \mathbf{1}_i\|_\infty + \|\mathbf{p}_t\|_\infty \leq B_i + \sqrt{k}\delta 20(B \vee \varepsilon).$$

Hence for each coordinate $i \in S$ we can apply the first part of Lemma B.12 so that after another $t = \frac{15}{32\eta w_{\min}^\star}\log\frac{w_{\min}^\star}{5\varepsilon}$ iterations we are done.

Hence the total number of required iterations is at most $t \leq \frac{45}{32\eta w_{\min}^\star}\log\frac{w_{\min}^\star}{\varepsilon}$.

$\qquad\square$

# C  Missing Proofs from Section A.3

This section provides proofs for the technical lemmas stated in section A.3.

## C.1  Proof of Lemma A.1

Looking at the updates given by equation 4 in appendix A.1 we have

$$\mathbf{1}_{S^c} \odot \mathbf{e}_{t+1} = \mathbf{1}_{S^c} \odot \mathbf{w}_t \odot \left(\mathbf{1} - 4\eta(\mathbf{s}_t - \mathbf{w}^\star + \mathbf{b}_t + \mathbf{p}_t)\right)^2 \tag{18}$$

$$= \mathbf{1}_{S^c} \odot \mathbf{e}_t \odot \left(\mathbf{1}_{S^c} - \mathbf{1}_{S^c} \odot 4\eta(\mathbf{s}_t - \mathbf{w}^\star + \mathbf{b}_t + \mathbf{p}_t)\right)^2 \tag{19}$$

$$= \mathbf{1}_{S^c} \odot \mathbf{e}_t \odot \left(\mathbf{1} - 4\eta(\mathbf{b}_t + \mathbf{p}_t)\right)^2 \tag{20}$$

and hence

$$\|\mathbf{1}_{S^c} \odot \mathbf{e}_{t+1}\|_\infty \le \|\mathbf{1}_{S^c} \odot \mathbf{e}_t\|_\infty \left(1 + 4\eta(\|\mathbf{b}_t\|_\infty + \|\mathbf{p}_t\|_\infty)\right)^2$$

which completes the proof for $\mathbf{1}_{S^c} \odot \mathbf{e}_t$.

On the other hand, Lemma B.16 deals with $\mathbf{1}_S \odot \mathbf{e}_t$ immediately and we are done.  □

## C.2  Proof of Lemma A.2

Note that

$$1 + 4\eta b_t \le 1 + 4\eta B$$

and hence

$$x_t \le x_0(1 + 4\eta B)^{2t}.$$

To ensure that $x_t \le \sqrt{x_0}$ it is enough to ensure that the right hand side of the above expression is not greater than $\sqrt{x_0}$. This is satisfied by all $t$ such that

$$t \le \frac{1}{2} \frac{\log \frac{1}{\sqrt{x_0}}}{\log(1 + 4\eta B)}$$

Now by using $\log x \le x - 1$ we have

$$\frac{1}{2} \frac{\log \frac{1}{\sqrt{x_0}}}{\log(1 + 4\eta B)} \ge \frac{1}{2} \frac{\log \frac{1}{\sqrt{x_0}}}{4\eta B}$$

$$= \frac{1}{32\eta B} \log \frac{1}{x_0^2}$$

which concludes our proof.  □

## C.3  Proof of Lemma A.3

For any index set $S$ of size $k+1$ let $\mathbf{X}_S$ be the $n \times (k+1)$ sub-matrix of $\mathbf{X}$ containing columns indexed by $S$. Let $\lambda_{\max}\left(\frac{1}{n}\mathbf{X}_S^\mathsf{T}\mathbf{X}_S\right)$ and $\lambda_{\min}\left(\frac{1}{n}\mathbf{X}_S^\mathsf{T}\mathbf{X}_S\right)$ denote the maximum and minimum eigenvalues of $\left(\frac{1}{n}\mathbf{X}_S^\mathsf{T}\mathbf{X}_S\right)$ respectively. It is then a standard consequence of the $(k+1, \delta)$-RIP that

$$1 - \delta \le \lambda_{\min}\left(\frac{1}{n}\mathbf{X}_S^\mathsf{T}\mathbf{X}_S\right) \le \lambda_{\max}\left(\frac{1}{n}\mathbf{X}_S^\mathsf{T}\mathbf{X}_S\right) \le 1 + \delta.$$

Let $\mathbf{z} \in \mathbb{R}^d$ be any $k$-sparse vector. Then, for any $i \in \{1, \ldots, d\}$ the joint support of $\mathbf{1}_i$ and $\mathbf{z}$ is of size at most $k + 1$. We denote the joint support by $S$ and we will also denote by $\mathbf{z}_S$ and $(\mathbf{1}_i)_S$ the restrictions of $\mathbf{z}$ and $\mathbf{1}_i$ on their support, i.e., vectors in $\mathbb{R}^{k+1}$. Letting $\|\cdot\|$ be the spectral norm, we

have

$$\left| \left( \frac{1}{n} \mathbf{X}^\mathsf{T} \mathbf{X} \mathbf{z} \right)_i - \mathbf{z}_i \right| = \left| \left\langle \frac{1}{n} \mathbf{X}^\mathsf{T} \mathbf{X} \mathbf{z}, \mathbf{1}_i \right\rangle - \langle \mathbf{z}, \mathbf{1}_i \rangle \right|$$

$$= \left| \left\langle \frac{1}{\sqrt{n}} \mathbf{X} \mathbf{z}, \frac{1}{\sqrt{n}} \mathbf{X} \mathbf{1}_i \right\rangle - \langle \mathbf{z}, \mathbf{1}_i \rangle \right|$$

$$= \left| \left\langle \frac{1}{\sqrt{n}} \mathbf{X}_S \mathbf{z}_S, \frac{1}{\sqrt{n}} \mathbf{X}_S (\mathbf{1}_i)_S \right\rangle - \langle \mathbf{z}_S, (\mathbf{1}_i)_S \rangle \right|$$

$$= \left| \left\langle \left( \frac{1}{n} \mathbf{X}_S^\mathsf{T} \mathbf{X}_S - \mathbf{I} \right) \mathbf{z}_S, (\mathbf{1}_i)_S \right\rangle \right|$$

$$\leq \left\| \frac{1}{n} \mathbf{X}_S^\mathsf{T} \mathbf{X}_S - \mathbf{I} \right\| \|\mathbf{z}\|_2 \|\mathbf{1}_i\|_2$$

$$\leq \delta \|\mathbf{z}\|_2$$

where the penultimate line follows by the Cauchy-Schwarz inequality and the last line follows by the $(k+1, \delta)$-RIP. Since $i$ was arbitrary it hence follows that

$$\left\| \left( \frac{1}{n} \mathbf{X}^\mathsf{T} \mathbf{X} - \mathbf{I} \right) \mathbf{z} \right\|_\infty \leq \delta \|\mathbf{z}\|_2 \leq \delta \sqrt{k} \|\mathbf{z}\|_\infty .$$

$\square$

## C.4  Proof of Lemma A.4

For any $i \in \{1, \ldots, d\}$ we can write $\mathbf{X}_i = \mathbf{X} \mathbf{1}_i$. The result is then immediate by the $(k+1, \delta)$-RIP since

$$\left\| \frac{1}{\sqrt{n}} \mathbf{X} \mathbf{1}_i \right\|_2^2 \leq (1+\delta) \|\mathbf{1}_i\|_2^2 \leq 2.$$

By the Cauchy-Schwarz inequality we then have, for any $i, j \in \{1, \ldots, d\}$,

$$\left| \left( \frac{1}{n} \mathbf{X}^\mathsf{T} \mathbf{X} \right)_{i,j} \right| \leq \left\| \frac{1}{\sqrt{n}} \mathbf{X}_i \right\|_2 \left\| \frac{1}{\sqrt{n}} \mathbf{X}_j \right\|_2 \leq 2$$

and for any $\mathbf{z} \in \mathbb{R}^d$ it follows that

$$\left\| \frac{1}{n} \mathbf{X}^\mathsf{T} \mathbf{X} \mathbf{z} \right\|_\infty \leq 2d \|\mathbf{z}\|_\infty .$$

$\square$

## C.5  Proof of Lemma A.5

Since for any column $\mathbf{X}_i$ of the matrix $\mathbf{X}$ we have $\|X_i\|_2 / \sqrt{n} \leq C$ and since the vector $\xi$ consists of independent $\sigma^2$-subGaussian random variables, the random variable $\frac{1}{\sqrt{n}} \left( \mathbf{X}^\mathsf{T} \xi \right)_i$ is $C^2 \sigma^2$-subGaussian.

It is then a standard result that for any $\varepsilon > 0$

$$\mathbb{P} \left( \left\| \frac{1}{\sqrt{n}} \mathbf{X}^\mathsf{T} \xi \right\|_\infty > \varepsilon \right) \leq 2d e^{-\frac{\varepsilon^2}{2C^2 \sigma^2}} .$$

Setting $\varepsilon = 2\sqrt{2C^2 \sigma^2 \log(2d)}$ we have with probability at least $1 - \frac{1}{8d^3}$ we have

$$\left\| \frac{1}{\sqrt{n}} \mathbf{X}^\mathsf{T} \xi \right\|_\infty \leq 4\sqrt{C^2 \sigma^2 \log(2d)}$$

$$\lesssim \sqrt{\sigma^2 \log d}.$$

$\square$

# D   Proof of Theorem 2

Recall the updates equations for our model parameters given in equations (3) and (4) as defined in Appendix A.1.

Since $\mathbf{w}_0 = 0$ we can rewrite the first update written on $\mathbf{u}$ and $\mathbf{v}$ as

$$
\begin{aligned}
\mathbf{u}_1 &= \mathbf{u}_0 \odot \left( 1 - 4\eta \left( -\mathbf{w}^\star + \left( \mathbf{I} - \frac{1}{n}\mathbf{X}^\mathsf{T}\mathbf{X} \right) \mathbf{w}^\star - \frac{1}{n}\mathbf{X}^\mathsf{T}\xi \right) \right), \\
\mathbf{v}_1 &= \mathbf{v}_0 \odot \left( 1 + 4\eta \left( -\mathbf{w}^\star + \left( \mathbf{I} - \frac{1}{n}\mathbf{X}^\mathsf{T}\mathbf{X} \right) \mathbf{w}^\star - \frac{1}{n}\mathbf{X}^\mathsf{T}\xi \right) \right).
\end{aligned}
\tag{21}
$$

By Lemma A.3 we have $\left\| \left( \mathbf{I} - \frac{1}{n}\mathbf{X}^\mathsf{T}\mathbf{X} \right)\mathbf{w}^\star \right\|_\infty \leq \frac{1}{20}w_{\max}^\star$. The term $\frac{1}{n}\mathbf{X}^\mathsf{T}\xi$ can be simply bounded by $\left\| \frac{1}{n}\mathbf{X}^\mathsf{T}\xi \right\|_\infty$. If $w_{\max}^\star \geq 5 \left\| \frac{1}{n}\mathbf{X}^\mathsf{T}\xi \right\|_\infty$ (note that otherwise returning a 0 vector is minimax-optimal) then

$$
\frac{1}{20}w_{\max}^\star + \left\| \frac{1}{n}\mathbf{X}^\mathsf{T}\xi \right\|_\infty \leq \frac{1}{4}w_{\max}^\star.
$$

We can hence bound the below term appearing in equation (21) as follows:

$$
\frac{3}{4}w_{\max}^\star \leq \left\| -\mathbf{w}^\star + \left( \mathbf{I} - \frac{1}{n}\mathbf{X}^\mathsf{T}\mathbf{X} \right)\mathbf{w}^\star - \frac{1}{n}\mathbf{X}^\mathsf{T}\xi \right\|_\infty \leq \frac{5}{4}w_{\max}^\star
$$

The main idea here is that we can recover the above factor by computing one gradient descent iteration and hence we can recover $w_{\max}^\star$ up to some multiplicative constants.

In fact, with $0 < \eta \leq \frac{1}{5w_{\max}^\star}$ so that the multiplicative factors are non-negative, the above inequality implies that

$$
1 + 3\eta w_{\max}^\star \leq f_{\max} \leq 1 + 5\eta w_{\max}^\star
$$

and so

$$
w_{\max}^\star \leq \frac{f_{\max} - 1}{3\eta} \leq \frac{5}{3}w_{\max}^\star
$$

which is what we wanted to show.

Note that after an application of this theorem we can now reset the step size to

$$
\frac{3\eta}{20\left( f_{\max} - 1 \right)}.
$$

This new step size satisfies the conditions of Theorems 1 and 3 while being at most two times smaller than required.

# E   Proof of Theorem 3

For proving Theorem 3 we first prove Propositions 3 and 4 which correspond to Propositions 1 and 2 but allows for different step sizes along each dimension. We present the proof of Proposition 3 in Section E.1.

**Proposition 3.** *Consider the setting of Proposition 1 and run Algorithm 2 with $\tau = 640$.*

*Then, for some early stopping time $T = O\left( \log \frac{w_{\max}^\star}{\zeta} \log \frac{1}{\alpha} \right)$ and any $0 \leq t \leq T$ we have*

$$
\begin{aligned}
\| \mathbf{s}_T - \mathbf{w}^\star \|_\infty &\leq \zeta, \\
\| \mathbf{e}_t \|_\infty &\leq \alpha.
\end{aligned}
$$

*Further, let $\eta_{T,j}$ be the step size for the $j^{th}$ coodinate at time $T$. Then, for all $j$ such that $|w_j^\star| > \zeta$ we have*

$$
\frac{1}{16} \cdot \frac{1}{20\left| w_j^\star \right|} \leq \eta_{T,j} \leq \frac{1}{20\left| w_j^\star \right|}.
$$

**Proposition 4.** *Consider the setting of updates given in equations (3) and (4). Fix any $\varepsilon > 0$ and suppose that the error sequences $(\mathbf{b}_t)_{t\geq 0}$ and $(\mathbf{p}_t)_{t\geq 0}$ satisfy for any $t \geq 0$:*

$$\|\mathbf{b}_t \odot \mathbf{1}_i\|_\infty \leq B_i \leq \frac{1}{10} w^\star_{\min},$$

$$\|\mathbf{p}_t\|_\infty \leq \frac{1}{20} \|\mathbf{s}_t - \mathbf{w}^\star\|_\infty.$$

*Suppose that*

$$\|\mathbf{s}_0 - \mathbf{w}^\star\|_\infty \leq \frac{1}{5} w^\star_{\min}.$$

*For each $i \in S$ let the step size satisfy $\frac{1}{\eta_i |w^\star_i|} \leq 320$. Then for all $t \geq 0$*

$$\|\mathbf{s}_t - \mathbf{w}^\star\|_\infty \leq \frac{1}{5} w^\star_{\min}$$

*and for any $t \geq 450 \log \frac{w^\star_{\min}}{\varepsilon}$ we have for any $i \in S$.*

$$|s_{t,i} - w^\star_i| \lesssim \delta \sqrt{k} \max_{j \in S} B_j \vee B_i \vee \varepsilon$$

*Proof.* We follow the same strategy as in the proof of Proposition 2. The only difference here is that the worst case convergence time $\frac{1}{\eta w^\star_{\min}}$ is replaced by $\max_{i \in S} \frac{1}{\eta_i |w^\star_i|} \leq 320$ and the result follows. $\qquad\square$

*Proof of Theorem 3.* The proof is identical to the proof of Theorem 1 with application of Proposition 1 replaced with Proposition 3 and in the easy setting the application of Proposition 2 replaced with an application of Proposition 4.

The only difference is that extra care must be taken when applying Proposition 4. First, note that the pre-conditions on step sizes are satisfied by Proposition 3. Second, the number of iterations required by Proposition 4 is fewer than step-size doubling intervals, and hence the step sizes will not change after the application of Proposition 3. In particular, Proposition 3 requires $450 \log \frac{w^\star_{\min}}{\varepsilon}$ iterations and we double the step sizes every $640 \log \frac{1}{\alpha}$ iterations. This finishes our proof. $\qquad\square$

## E.1 Proof of Proposition 3

Recall the proof of Proposition 1 that we have shown in Appendix B.5. We have used a constant step size $\eta \leq \frac{5}{96 w^\star_{\max}}$. With a constant step size this is in fact unavoidable up to multiplicative constants – for larger step sizes the iterates can explode.

Looking at our proof by induction of Lemma B.17, the inefficiency of Algorithm 1 comes from doubling the number of iterations during each induction step. This happens because during the $i^{th}$ induction step the smallest coordinates of $\mathbf{w}^\star$ that we consider are of size $2^{-i-1} w^\star_{\max}$. For such coordinates, step size $\eta \leq \frac{5}{96 w^\star_{\max}}$ could be at least $2^i$ times bigger and hence the convergence would be $2^i$ times faster. The lemmas derived in Appendix B.2 indicate that fitting signal of such size will require number of iterations proportional to $\frac{1}{\eta 2^{-i-1} w^\star_{\max}} = 2^{i+1} \frac{1}{\eta w^\star_{\max}}$ which is where the exponential increase in the number of iterations for each induction step comes from.

We can get rid of this inefficiency if for each coordinate $j$ we use a different step size, so that for all $j$ such that $|w^\star_j| \ll w^\star_{\max}$ we set $\eta_j \gg \frac{5}{96 w^\star_{\max}}$. In fact, the only constraint we have is that $\eta_j$ never exceeds $\frac{5}{96 |w^\star|_j}$. To see that we can change the step sizes for small enough signal in practice, note that after two induction steps in Proposition 1 we have $\|\mathbf{s}_t - \mathbf{w}^\star\|_\infty \leq \frac{1}{4} w^\star_{\max}$ and $\|\mathbf{e}_t\|_\infty \leq \alpha$. We can then show, that for each $j$ such that $|w^\star|_j > \frac{1}{2} w^\star_{\max}$ we have $|w_{t,j}| > \frac{1}{4} w^\star_{\max}$. On the other hand, if $|w^\star|_j \leq \frac{1}{8} w^\star_{\max}$ then $w_{t,j} \leq w^\star_j + 4B_1 \leq \frac{1}{4} w^\star_{\max}$, where $B_1$ is given as in Lemma B.17. In particular, after the second induction step one can take all $j$ such that $|w_{t,j}| \leq \frac{1}{4} w^\star_{\max}$ and double its associated step sizes.

We exploit the above idea in the following lemma, which is a counterpart to Lemma B.17. One final thing to note is that we do not really know what $w^\star_{\max}$ is which is necessary in the argument sketched

above. However, in Theorem 2 we showed that we can compute some $\hat{z}$ such that $w_{\max}^\star \le \hat{z} \le 2w_{\max}^\star$ and as we shall see this is enough.

**Lemma E.19** (Counterpart to Lemma B.17 with increasing step sizes). *Consider the same setting of Lemma B.17. Run Algorithm 2 with $\tau = 640$ and parametrization $\mathbf{w}_t = \mathbf{u}_t \odot \mathbf{u}_t$.*

*Then, for $t = \left\lceil 640 \log_2 \frac{w_{\max}^\star}{\zeta} \log \frac{1}{\alpha} \right\rceil$ and any $j$ we have*

$$\left\| \mathbf{w}_t - \mathbf{w}_+^\star \right\|_\infty \le \zeta$$

$$\alpha^2 \prod_{i=0}^{t-1} (1 + 4\eta_{t,j}(\|\mathbf{b}_t\|_\infty + \|\mathbf{p}_t\|_\infty))^2 \le \alpha.$$

*Proof.* Following the notation used in Lemma B.17 for any integer $i \ge -1$ let $T_i := T$ and $\bar{T}_i := \sum_{j=0}^i T_j = (i+1)T$. We remark now that we have the same $T$ for each induction step in contrast to exponentially increasing number of iterations in Lemma B.17. Let $B_i := \frac{1}{40} 2^{-i} w_{\max}^\star$. Let $m = \left\lceil \log_2 \frac{w_{\max}^\star}{\zeta} \right\rceil$ so that $\gamma = \frac{C_\gamma}{m}$. We will prove our claim by induction on $i = 0, 1, \ldots, m-1$.

*Induction hypothesis for $i \in \{0, \ldots, m\}$*

1. For any $j < i$ and $\bar{T}_{j-1} \le t < \bar{T}_j$ we have $\left\| \mathbf{w}_t - \mathbf{w}_+^\star \right\|_\infty \le 2^{-j} w_{\max}^\star$. In particular, this induction hypothesis says that we halve the convergence distance during each induction step.

2. We have $\left\| \mathbf{w}_{\bar{T}_{i-1}} - \mathbf{w}_+^\star \right\|_\infty \le 2^{-i} w_{\max}^\star$. This hypothesis controls the convergence distance at the beginning of each induction step.

3. For any $j$ such that $w_j^\star \le 20 B_i = 2^{-i-1} w_{\max}^\star$ we have $\alpha^3 \le w_{\bar{T}_{i-1},j} \le w_j^\star + 4 B_i$. On the other hand, for any $j$ such that $w_j^\star \ge 20 B_i$ we have $\alpha^3 \le w_{\bar{T}_{i-1},j} \le \frac{6}{5} w_j^\star$.

4. Let $l$ be any integer such that $0 \le l \le i$. Then for any $j$ such that $2^{-l-1} w_{\max}^\star < w_j^\star \le 2^{-l} w_{\max}^\star$ we have
$$2^{l-3} \eta_{0,j} \le \eta_{\bar{T}_{i-1},j} \le 2^l \eta_{0,j}$$
For any $j$ such that $w_j^\star \le 2^{-i-1}$ we have
$$2^{i-2} \eta_{0,j} \le \eta_{\bar{T}_{i-1},j} \le 2^{(i-1)\vee 0} \eta_{0,j}.$$

   In particular, the above conditions ensure that we $\eta_{t,j}$ never exceeds $\frac{1}{20 w_j^\star}$ so that the step-size pre-conditions of all lemmas derived in previous appendix sections always hold during each induction step. Further, it ensures that once we fit small coordinates, the step size is up to absolute constants as big as possible.

We remark the that in addition to induction hypotheses used in Lemma B.17 the fourth induction hypothesis allows to control what happens to the step sizes with our doubling step size scheme. There is also a small modification to the third induction hypothesis, where right now we sometimes allow $w_{t,j} > w_j^\star + 4 B_i$ because due to increasing step sizes we have to deal iterates larger than target slightly differently. In particular, we can only apply Lemma B.14 for coordinates $j$ with sufficiently small $w_j^\star$, because the step sizes of such coordinates will be larger which allows for faster convergence.

*Base case*
For $i = 0$ all conditions hold since for all $j$ we have $0 \le \alpha^2 = w_{0,j} < w_j^\star$ and since all $\eta_{0,j} \le \frac{1}{20 w_{\max}^\star}$.

*Induction step*
Assume that the induction hypothesis holds for some $0 \le i < m$. We will show that it also holds for $i + 1$.

1. The proof is based on monotonic convergence to $B_i$ tube argument and is identical to the one used in Lemma B.17 with the same conditions on $C_b$ and $C_\gamma$.

2. Similarly to the proof of Lemma B.17 here we only need to handle coordinates $j$ such that $w_j^\star > 20B_i = 2^{-i-1}w_{\max}^\star$ and $\left|w_{\bar{T}_{i-1},j} - w_j^\star\right| > 2^{-i-1}w_{\max}^\star$.

   If $w_{\bar{T}_{i-1},j} \leq w_j^\star$ we apply the second part of Lemma B.12 with $\varepsilon = 19B_i$ to obtain that for any

   $$t \geq \frac{1}{2}\frac{1}{\eta_{\bar{T}_{i-1},j}w_j^\star}\log\frac{1}{\alpha^4}$$
   $$\geq \frac{15}{32\eta w_j^\star}\frac{1}{\eta_{\bar{T}_{i-1},j}w_j^\star}\log\frac{(w_j^\star)^2}{w_{\bar{T}_{i-1},j}\cdot 19B_i}$$

   iterations the following holds

   $$\left|w_{\bar{T}_{i-1}+t,j} - w_j^\star\right| \leq 20B_i \leq 2^{-i-1}w_{\max}^\star.$$

   By the fourth induction hypothesis and by definition of $\eta_{0,j}$ we have

   $$\frac{1}{\eta_{\bar{T}_{i-1},j}w_j^\star} \leq \frac{8}{\eta_{0,j}w_{\max}^\star} \leq 16\cdot 20.$$

   and hence $T$ iterations are enough.

   If $w_{\bar{T}_{i-1},j} \geq w_j^\star$ by the third induction hypothesis we also have $w_{\bar{T}_{i-1},j} \leq \frac{6}{5}w_j^\star$ so that the pre-condition of Lemma B.11 apply and we are done, since it requires fewer iterations than considered above.

3. We first deal with the upper-bound. For $j$ such that $w_j^\star \geq 20B_i$ we have by the third induction hypothesis $w_{\bar{T}_{i-1},j} \leq \frac{6}{5}w_j^\star$ and hence by the monotonic convergence to $B_i$-tube argument given in Lemma B.10 this bound still holds after the $i^{th}$ induction step. For any $j$ such that $w_j^\star \leq 20B_i$ we use Lemma B.14 and the fourth induction hypothesis $\eta_{\bar{T}_{i-1},j} \geq 2^{i-3}\eta_{0,j}$ to show that after

   $$T \geq \frac{32}{\eta_{0,j}w_{\max}^\star} \geq \frac{2^{i+2}}{\eta_{\bar{T}_{i-1},j}w_{\max}^\star} = \frac{1}{10\eta_{\bar{T}_{i-1},j}B_i}.$$

   iterations for any such $j$ we have $w_{\bar{T}_{i-1}+t} \leq w_j^\star + 2B_i = w_j^\star + 4B_{i+1}$. Finally, this implies that if $10B_i \leq w_j^\star \leq 20B_i$ then after $T$ iterations $w_{\bar{T}_i,j} \leq \frac{6}{5}w_j^\star$.

   To prove the lower-bound, note that during the $i^{th}$ induction step for any $j$ we have $\eta_{j,\bar{T}_{i-1}} \leq 2^i\eta_{0,j}$ since each step size at most doubles after every induction step. Hence during the $i^{th}$ induction step, the accumulation of error can be upper-bounded by

   $$\prod_{i=\bar{T}_{i-1}}^{\bar{T}_i-1}(1+4\eta_{\bar{T}_{i-1},j}(\|\mathbf{b}_i\|_\infty + \|\mathbf{p}_i\|_\infty))^2$$
   $$\leq (1+4\cdot 2^i\eta_{0,j}(\|\mathbf{b}_i\|_\infty + \|\mathbf{p}_i\|_\infty))^{2T}$$
   $$\leq (1+4\cdot\eta_{0,j}(\|\mathbf{b}_i\|_\infty + \|\mathbf{p}_i\|_\infty))^{2\cdot 2^i T}.$$

   Now since our $2_iT$ is simply the same $T_i$ as used in Lemma B.17 rescaled at most 8 times, the same bounds holds on the accumulation of error as in Lemma B.17 with absolute constants $C_b$ and $C_\gamma$ rescaled by $\frac{1}{8}$ in this lemma. This completes the third induction hypothesis step.

4. After the $i^{th}$ induction step (recall that the induction steps are numbered starting from 0), if $i \geq 1$ our step size scheme doubles $\eta_{\bar{T}_i,j}$ if $w_{\bar{T}_i,j} \leq 2^{-i-2}\hat{z}$. Recall that after $i^{th}$ induction step we have $\|\mathbf{w}_t - \mathbf{w}_+^\star\|_\infty \leq 2^{-i-1}w_{\max}^\star$.

   For every $j$ such that $w_j^\star > 2^{-i}w_{\max}^\star$ we have $w_{\bar{T}_i,j} > 2^{-i-1}w_{\max}^\star \geq 2^{-i-2}\hat{z}$ and hence $\eta_{\bar{T}_i,j}$ will not be affected.

For every $j$ such that $w_j^\star \leq 2^{-i-3} w_{\max}^\star$ we have $w_{\bar{T}_i, j} \leq w_j^\star + 4B_{i+1} \leq 2^{-i-2} w_{\max}^\star$ and for such $j$ the step size will be doubled.

Hence for any non-negative integer $k$ and any $j$ such that $2^{-k-1} w_{\max}^\star < w_j^\star \leq 2^{-k} w_{\max}^\star$ the corresponding step size will be doubled after $i^{th}$ induction step for $i = 1, \ldots, k - 3$ and will not be touched anymore after and including the $k + 1^{th}$ induction step. We are only uncertain about what happens for such $j$ after the $k - 2, k - 1$ and $k^{th}$ induction steps, which is where the factor of $8$ comes from. This concludes the proof of the fourth induction hypothesis.

The result then follows after $mT$ iterations which is what we wanted to show. $\qquad\square$

Similarly to the proof of Proposition 1 we can extend the above Lemma to a general setting (i.e. parametrization $\mathbf{w}_t := \mathbf{u}_t \odot \mathbf{u}_t - \mathbf{v}_t \odot \mathbf{v}_t$) by using Lemma B.16. The following proposition then corresponds to Proposition 1 but allows to use our increasing step sizes scheme.

*Proof of Proposition 3.* Immediate by Lemma B.16 by the same argument as used in the proof of Proposition 1. $\qquad\square$

## F   Gradient Descent Updates

We add the derivation of gradient descent updates for completeness. Let $\mathbf{w} = \mathbf{u} \odot \mathbf{u} - \mathbf{v} \odot \mathbf{v}$ and suppose

$$\mathcal{L}(\mathbf{w}) = \frac{1}{n} \|\mathbf{X}\mathbf{w} - \mathbf{y}\|_2^2 \,.$$

We then have for any $i = 1, \ldots, d$

$$
\begin{aligned}
\frac{\partial}{\partial u_i} \mathcal{L}(\mathbf{w}) &= \frac{1}{n} \sum_{j=1}^{n} \frac{\partial}{\partial u_i} (\mathbf{X}\mathbf{w} - \mathbf{y})_j^2 \\
&= \frac{1}{n} \sum_{j=1}^{n} 2(\mathbf{X}\mathbf{w} - \mathbf{y})_j \cdot \frac{\partial}{\partial u_i} (\mathbf{X}\mathbf{w} - \mathbf{y})_j \\
&= \frac{1}{n} \sum_{j=1}^{n} 2(\mathbf{X}\mathbf{w} - \mathbf{y})_j \cdot \frac{\partial}{\partial u_i} (\mathbf{X}(\mathbf{u} \odot \mathbf{u}))_j \\
&= \frac{1}{n} \sum_{j=1}^{n} 2(\mathbf{X}\mathbf{w} - \mathbf{y})_j \cdot 2u_i X_{ji} \\
&= 4u_i \frac{1}{n} \sum_{j=1}^{n} X_{ji} (\mathbf{X}\mathbf{w} - \mathbf{y})_j \\
&= 4u_i \frac{1}{n} \left(\mathbf{X}^\mathsf{T}(\mathbf{X}\mathbf{w} - \mathbf{y})\right)_i
\end{aligned}
$$

and hence

$$\nabla_{\mathbf{u}} \mathcal{L}(\mathbf{w}) = \frac{4}{n} \mathbf{X}^\mathsf{T}(\mathbf{X}\mathbf{w} - \mathbf{y}) \odot \mathbf{u},$$

$$\nabla_{\mathbf{v}} \mathcal{L}(\mathbf{w}) = -\frac{4}{n} \mathbf{X}^\mathsf{T}(\mathbf{X}\mathbf{w} - \mathbf{y}) \odot \mathbf{v}.$$

## G   Comparing Assumptions to [30]

We compare our conditions on $\alpha, \delta$ and $\eta$ to the related work analyzing implicit regularization effects of gradient descent for noiseless low-rank matrix recovery problem with a similar parametrization [30].

The parameter $\alpha$ plays a similar role in both papers: $\ell_2$ (or reconstruction) error in the noiseless setting is directly controlled by the size of $\alpha$ as we show in Corollary 1. In both settings the number of iterations is affected only by a multiplicative factor of $O(\log 1/\alpha)$.

The conditions imposed on $\alpha$ and $\eta$ in [30] are much stronger than required in our work. Our results do not follow from the main result of [30] by considering a matrix recovery problem for the ground truth matrix $\mathrm{diag}(\mathbf{w}^\star)$. Letting $\kappa = w_{\max}^\star/w_{\min}^\star$ the assumptions of [30] require $\delta \lesssim 1/(\kappa^3\sqrt{k}\log^2 d)$ and $\eta \lesssim \delta$ yielding $\Omega(\kappa/\eta \log 1/\alpha) = \Omega(\kappa^4 \log^2 d\sqrt{k}\log 1/\alpha)$ iteration complexity. In contrast, our theorem only requires $\delta$ to scale only as $1/\log\kappa$. We are able to set the step-size using data and do not rely on knowing the unknown quantities $\kappa$ and $k$.

Crucially, when $w_{\min}^\star \lesssim \|\mathbf{X}^\mathsf{T}\xi\|_\infty/n$ in the sub-Gaussian noise setting the assumption $\delta \lesssim 1/(\kappa^3\sqrt{k}\log^2 d)$ implies that for sample size $n$, the RIP parameter $\delta = O(n^{-3/2})$, which is in general impossible to satisfy, e.g. when the entries of $\mathbf{X}$ are i.i.d. Gaussian. Hence moving the dependence on $\kappa$ into a logarithmic factor as done in our analysis is key for handling the general noisy setting. For this reason, our proof techniques are necessarily quite different and may be of independent interest.

## H  Comparing Our Results to [56]

Instead of using parametrization $w = u \odot u - v \odot v$, the authors of [56] consider a closely related Hadamard product reparametrization $w = u \odot v$ and perform gradient descent updates on $u$ and $v$ for the least squares objective function with no explicit regularization. This work is related to ours in that the ideas of implicit regularization and sparsity are combined to yield a statistically optimal estimator for sparse recovery under the RIP assumption. In this section, we compare this work to ours, pointing out the key similarities and differences.

To simplify the notation, in all points below we assume that $w_{\min}^\star \gtrsim \|\mathbf{X}^\mathsf{T}\xi\|_\infty/n$ so that the variable $m$ used in [56] coincides with $w_{\min}^\star$ used in this paper.

**(Difference) Properly handling noisy setting:**  Let $\kappa := w_{\max}^\star/w_{\min}^\star$. The assumption (B) in [56] requires $\mathbf{X}/\sqrt{n}$ to satisfy $(k+1,\delta)$-RIP with $\delta \lesssim \frac{1}{\kappa\sqrt{k}\log(d/\alpha)}$. On the other hand, for our results to hold it is enough to have $\delta \lesssim \frac{1}{\sqrt{k}\log\kappa}$. Moving $\kappa$ into a logarithmic factor is the key difference, which requires a different proof technique and also allows to handle the noise properly. To see why the latter point is true, consider $w_{\min}^\star \asymp \sigma\sqrt{\log d}/\sqrt{n}$. The assumption (B) in [56] then requires $\delta = O(1/(\sqrt{k}\sqrt{n}))$, which is in general impossible to satisfy with random design matrices, e.g., when entries of $\mathbf{X}$ are i.i.d. Gaussian. Hence, in contrast to our results, the results of [56] cannot recover the smallest possible signals (i.e., $\mathbf{w}^\star$ coordinates of order $\sigma\sqrt{\log d}/\sqrt{n}$).

**(Difference) Computational optimality:**  In this paper we consider an increasing step size scheme which yields up to poly-logarithmic factors a computationally optimal algorithm for sparse recovery under the RIP. On the other hand, only constant step sizes were considered in [56], which does not result in a computationally optimal algorithm.

Moreover, due to different constraints on step sizes, the two papers yield different iteration complexities for early stopping times even in the setting of running gradient descent with constant step sizes. In [56, Theorem 3.2] the required number of iterations is $\Omega(\frac{\log(d/\alpha)}{\eta w_{\min}^\star}) = \Omega(\frac{\kappa}{w_{\min}^\star}\log^2(d/\alpha))$. If $w_{\min}^\star \asymp \sigma\sqrt{\log d}/\sqrt{n}$ the required number of iterations is then $\Omega(\frac{n w_{\max}^\star}{\sigma^2}\log(d/\alpha))$. On the other hand, in our paper Theorem 1 together with step size tuned by using Theorem 2, requires $O(\kappa\log\alpha^{-1}) = O(\frac{\sqrt{n}w_{\max}^\star}{\sigma}\log\alpha^{-1})$ iterations, yielding an algorithm faster by a factor of $\sqrt{n}$.

**(Difference) Conditions on step size:**  We require $\eta \lesssim 1/w_{\max}^\star$ while [56] requires (Assumption (C)) that $\eta \lesssim \frac{w_{\min}^\star}{w_{\max}^\star}(\log\frac{d}{\alpha})^{-1}$. The crucial difference is that this step size can be much smaller than $1/w_{\max}^\star$ required in our theorems and impacts computational efficiency as discussed in the computational optimality paragraph above.

Furthermore, a crucial result in our paper is Theorem 2 which allows us to optimally tune the step size with an estimate of $w_{\max}^\star$ that can be computed from the data. On the other hand, in [56] $\eta$

also depends on $w_{\min}^\star$. It is not clear how to choose such an $\eta$ in practice and hence it becomes an additional hyperparameter which needs to be tuned.

**(Difference) Dependence on $w_{\max}^\star$:** Our results establish explicit dependence on $w_{\max}^\star$, while assumption (A) in [56] requires $w_{\max}^\star \lesssim 1$.

**(Similarity) Recovering only coordinates above the noise level:** In both papers, the early stopping procedure stops while for all $i \in S$ such that $|w_i^\star| \lesssim \|\mathbf{X}^\mathsf{T}\xi\|_\infty/n$ we have $w_{t,i} \approx 0$. Essentially, such coordinates are treated as if they did not belong to the true support, since they cannot be recovered as certified by minimax-optimality bounds.

**(Similarity) Statistical optimality:** Both papers achieve minimax-optimal rates with early stopping and also prove dimension-independent rates when $w_{\min}^\star \gtrsim \|\mathbf{X}^\mathsf{T}\xi\|_\infty/n$. Our dimension-independent rate (Corollary 3) has an extra $\log k$ not present in results of [56]. We attribute this difference to stronger assumptions imposed on RIP parameter $\delta$ in [56]. Indeed, the $\log k$ factor comes from the $\delta\sqrt{k}\,\|\mathbf{X}^\mathsf{T}\xi/n \odot \mathbf{1}_S\|_\infty$ term in Theorems 1 and 3, which gets smaller with decreasing $\delta$.

# I  Further Improvements

In this section we expand on the potential improvements of our work outlined in Section 6.

**Sub-Optimal Sample Complexity.** Our RIP parameter $\delta$ scales as $\widetilde{O}(1/\sqrt{k})$. We remark that such scaling on $\delta$ is less restrictive than in [30, 56] (see Appendix G and H). If we consider, for example, sub-Gaussian isotropic designs, then satisfying such an assumption requires $n \gtrsim k^2 \log(ed/k)$ samples. To see that, consider an $n \times k$ i.i.d. standard normal ensemble which we denote by $\mathbf{X}$. By standard results in random-matrix theory [50, Chapter 6], $\|\mathbf{X}^\mathsf{T}\mathbf{X}/n - \mathbf{I}\| \lesssim \sqrt{k/n} + k/n$ where $\|\cdot\|$ denotes the operator norm. Hence, we need $n \gtrsim k^2$ to satisfy $\|\mathbf{X}^\mathsf{T}\mathbf{X}/n - \mathbf{I}\| \lesssim 1/\sqrt{k}$.

Note that Theorems 1 and 3 provide coordinate-wise bounds which is in general harder than providing $\ell_2$ error bounds directly. In particular, under the condition that $\delta = \widetilde{O}(1/\sqrt{k})$, our main theorems imply minimax-optimal $\ell_2$ bounds; this requirement on $\delta$ implies that $n$ needs to be at least quadratic in $k$. Hence we need to answer two questions. First, do we need sample complexity quadratic in $k$ to obtain minimax-rates? The left plot in Figure 7 suggests that linear sample complexity in $k$ is enough for our method to match and eventually exceed performance of the lasso in terms of $\ell_2$ error. Second, is it necessary to change our $\ell_\infty$ based analysis to an $\ell_2$ based analysis in order to obtain optimal sample complexity? The right plot in Figure 7 once again suggests that sample complexity linear in $k$ is enough for our main theorems to hold.

Figure 7: Sample complexity requirements. We let $d = 5000, \sigma = 1$ and $\mathbf{w}_S^\star = \mathbf{1}_S$. The plot on the left computes the $\log_2$ error ratio for our method (stopping time chosen by cross-validation) and the lasso ($\lambda$ chosen optimally using knowledge of $\mathbf{w}^\star$). The plot on the right computes $\|\mathbf{w}_t \odot \mathbf{1}_{S^c}\|_\infty$ for optimally chosen $t$.

**Relaxation to the Restricted Eigenvalue (RE) Assumption.** The RIP assumption is crucial for our analysis. However, the lasso satisfies minimax optimal rates under less restrictive assumptions, namely, the RE assumption introduced in [9]. The RE assumption with parameter $\gamma$ requires that $\|\mathbf{Xw}\|_2^2/n \geq \gamma\|\mathbf{w}\|_2^2$ for vectors $\mathbf{w}$ satisfying the cone condition $\|\mathbf{w}_{S^c}\|_1 \leq c\|\mathbf{w}_S\|_1$ for a suitable choice of constant $c \geq 1$. In contrast to RIP, RE only imposes constraints on the *lower* eigenvalue of $\mathbf{X}^\mathsf{T}\mathbf{X}/n$ for approximately sparse vectors and can be satisfied by random *correlated* designs [36, 42]. The RE condition was shown to be necessary for any polynomial-time algorithm returning a sparse vector and achieving fast rates for prediction error [55].

We sample i.i.d. Gaussian ensembles with covariance matrices equal to $(1-\mu)\mathbf{I} + \mu\mathbf{1}\mathbf{1}^\mathsf{T}$ for $\mu = 0$ and $0.5$. For $\mu = 0.5$ the RIP fails but the RE property holds with high probability [50, Chapter 7].

In Figure 8 we show empirically that our method achieves the fast rates and eventually outperforms the lasso even when we violate the RIP assumption.

Figure 8: Violating the RIP assumption. We consider the same setting as in Figure 3 with rows of $\mathbf{X}$ sampled from a Gaussian distribution with covariance matrix equal to $(1 - \mu)\mathbf{I} + \mu\mathbf{1}\mathbf{1}^{\mathsf{T}}$.

## J  Table of Notation

We denote vectors with boldface letters and real numbers with normal font. Hence $\mathbf{w}$ denotes a vector, while for example, $w_i$ denotes the $i^{th}$ coordinate of $\mathbf{w}$. We let $\mathbf{X}$ be a $n \times d$ design matrix, where $n$ is the number of observations and $d$ is the number of features. The true parameter is a $k$-sparse vector denoted by $\mathbf{w}^\star$ whose unknown support is denoted by $S \subseteq \{1, \ldots, d\}$. We let $w_{\max}^\star = \max_{i \in S} |w_i^\star|$ and $w_{\min}^\star = \min_{i \in S} |w_i^\star|$. We let $\mathbf{1}$ be a vector of ones, and for any index set $A$ we let $\mathbf{1}_A$ denote a vector equal to 1 for all coordinates $i \in A$ and equal to 0 everywhere else. We denote coordinate-wise product of vectors by $\odot$ and coordinate-wise inequalities by $\preccurlyeq$. With a slight abuse of notation we write $\mathbf{w}^2$ to mean coordinate-wise square of each element for a vector $\mathbf{w}$. Finally, we denote inequalities up to multiplicative absolute constants, meaning that they do not depend on any parameters of the problem, by $\lesssim$.

Table 1: Table of notation

| Symbol | Description |
| --- | --- |
| $n$ | Number of data points |
| $d$ | Number of features |
| $k$ | Sparsity of the true solution |
| $\mathbf{w}^\star$ | Ground truth parameter |
| $w_{\max}^\star$ | $\max_{i \in \{1,\ldots,k\}} |w_i^\star|$ |
| $w_{\min}^\star$ | $\min_{i \in \{1,\ldots,k\}} |w_i^\star|$ |
| $\kappa$ | $w_{\max}^\star / w_{\min}^\star$ |
| $\kappa^{\text{eff}}$ | $w_{\max}^\star / (w_{\min}^\star \vee \varepsilon \vee (\|\mathbf{X}^\mathsf{T} \xi\|_\infty / n))$ |
| $\odot$ | Coordinatewise multiplication operator for vectors |
| $\preccurlyeq$ | A coordinatewise inequality symbol for vectors |
| $\lesssim$ | An inequality up to some multiplicative absolute constant |
| $\mathbf{w}_t$ | Gradient descent iterate at time $t$ equal to $u_t \odot u_t + v_t \odot v_t$ |
| $\mathbf{u}_t$ | Parametrization of the positive part of $w_t$ |
| $\mathbf{v}_t$ | Parametrization of the negative part of $w_t$ |
| $\alpha$ | Initialization of $u_0$ and $v_0$ |
| $\eta$ | The step size for gradient descent updates |
| $\mathbf{w}_t^+$ | $u_t \odot u_t$ |
| $\mathbf{w}_t^-$ | $v_t \odot v_t$ |
| $S$ | Support of the true parameter $w^*$ |
| $S^+$ | Support of positive elements of the true parameter $w^*$ |
| $S^-$ | Support of negative elements of the true parameter $w^*$ |
| $\mathbf{1}_A$ | A vector with coordinates set to 1 on some index set $A$ and 0 everywhere else |
| $\mathbf{1}_i$ | A short-hand notation for $\mathbf{1}_{\{i\}}$ |
| $\mathbf{s}_t$ | The signal sequence equal to $\mathbf{1}_{S^+} \odot w_t^+ + \mathbf{1}_{S^-} \odot w_t^-$ |
| $\mathbf{e}_t$ | The error sequence equal to $\mathbf{1}_{S^c} \odot w_t + \mathbf{1}_{S^-} \odot w_t^+ + \mathbf{1}_{S^+} \odot w_t^-$ |
| $\mathbf{b}_t$ | Represents sequences of bounded errors |
| $\mathbf{p}_t$ | Represents sequences with errors proportional to the convergence distance $\|\mathbf{s}_t - \mathbf{w}^\star\|_\infty$ |