[Reviews · NeurIPS 2019]

Reviewer 1



The paper is well written, with emphasis to follow the main results and the implications. Some questions for clarification below: 1. What is the intuition for the choice of this reparametrization ? Is it the reparametrization which leads to sparse of the solutions or the choice of the free parameters ? 2. Though the authors mention the existing works on implicit L2 regularization, it would be interesting (and self-contained) to contrast the implicit L2 vs the L1 regularisation in terms of the algorithms and the results. 3. What are the possibilities of extensions of these sparse recovery towards structured sparse ones ? Minor comments / typos: 1. Algorithm 2, step 2. Duplicate notations for ‘m’, being used as a scaling factor (scalar) as well as a vector. 2. Line 181. Maximum noise term w*_max ? 3. Line 243. Constat -> constant

Reviewer 2



The paper's introduction is well written, the structure is also good. However, some important points could be made more clear, for example: 1. line 164-165: Where did you show the corresponding optimization path contains sparse solutions? Is the entire path sparse or it is only sparse at some iterations? It will help the reader if some more intuitions behind the algorithm could be provided. 2. Some notations in the main results should be clearly defined. For example, w*_{max} is the largest entry of w*. 3. The growth condition for dimension d with respect to n?

Reviewer 3



The results are very interesting, technically sound, and the paper is well written, in particular the proof sketch is very useful. The main critical comment I have is that the results require the RIP constant to satisfy \delta = 1/sqrt{k}, where k is the sparsity. A sub-Gaussian random matrix satisfies the RIP of oder k with high probability provided that the number of measurements (or rows of the matrix A \in \reals^{n\times d}), n, obey: n >= c \delta^{-2}(k log(d/s)) With \delta = 1/sqrt{k} as required by the theory, the number of measurements/rows of A needs to satisfy: n >= c k^2 log(d/s) I.e., the measurements need to be quadratic in the sparsity in contrast to l1 minimization, which only requires the RIP constant to be a fixed constant, independent of the sparsity, to succeed, and thus only requires the number of measurements to be slightly larger than linear. Thus, the algorithm presented (or at least its guarantees) are highly suboptimal in the sample complexity. It would be important to prominently mention that (otherwise the title `optimal sparse recovery' is misleading). Also it would be interesting to add some numerical studies that show what happens when this condition is violated, to investigate whether the condition is an artifact of the theory or a property of the algorithm. For example, one could do phase transition curves (different to the ones in the paper, relating recovery performance k versus n) and compare them to lasso.

[Author Response · NeurIPS 2019]

Before we begin, we highlight one of our key contributions: **Algorithm 2**. It showcases a remarkable *interplay between*
*statistics and optimization*: the *increasing* step sizes scheme (required for computational optimality) only works because
we rely on early stopping and do not aim to fully optimize the training objective. In contrast, most results in optimization
literature consider gradient descent with constant or *decreasing* step sizes to ensure convergence to the objective.
**Providing more intuition (R#1 and R#2).** For gradient descent (GD), our reparameterization turns *additive* updates
into *multiplicative* updates (see lines 69-74). As a result, the scale of the parameter can be understood as inertia – the
small parameters have a tendency to remain almost unchanged, while the larger parameters are more sensitive to the
gradient size with respect to the standard parameterization $\mathbf{w}$. Sparsity is induced with reparameterization *together* with
small initialization size (one without the other doesn't work). For more intuition see the proof sketch (Theorem 4),
simulations and appendices A and B. Finally, previous work in the literature shows that GD implicitly regularizes the $\ell_2$
norm. This corresponds to minimizing $\ell_1$ norm of $\mathbf{w}$ in our parameterization on $\mathbf{u}$, $\mathbf{v}$ (see lines 255-256).
**Choice of hyperparameters (R#1 and R#2).** As discussed in lines 201-208, we only need to know $w^\star_{\max}$ up to
multiplicative factors to properly initialize $\alpha$ and $\eta$. Theorem 2 shows how to obtain such an estimate. Hence we only
need to tune the stopping time, which can be done by cross-validation.

**1.** See our paragraphs on intuition and hyperparameters above.
**2.** Most work on implicit $\ell_2$ regularization focus on GD with a constant step size usually stopping at $\Theta(\sqrt{n})$ iterations;
$(\eta t)^{-1}$ corresponds to the Ridge regression $\lambda$. In lines 195-200 we discuss connections to Thm 1 and 3. On the other
hand, we are not aware of other work on implicit regularization achieving computational optimality via an increasing
step sizes scheme (Alg. 2). We highlight that in our case implicit regularizer is not strictly $\ell_1$ norm (see lines 125-126
and 334-339) and our work, to the best of our knowledge, is first to induce sparsity *implicitly* in a general noisy setting.
**3.** This remains an open question not considered in our paper. We believe that a good starting point would be to
experiment with individualized initialization sizes and step sizes among each dimension/group.
**Improvements section.** We agree with the suggestion and plan to add an additional paragraph to the related literature
section, expanding on the second point above. Subject to space considerations we will also expand on intuition.

**1.** For sparsity of the optimization path see proof sketch (Thm 4), simulations (lines 314-316)
and the main proofs. For sparsity at the stopping time, see the $\ell_\infty$ bound on $S^c$ in Thm 1.
**2.** $w^\star_{\max}$ is defined in line 84. Line 151 refers to table of notation.
**3.** Since $\mathbf{X}$ needs to only satisfy RIP, $n$ depends only logarithmically on $d$.
**Improvements section.** For intuition and hyperparameters, see the two paragraphs at the
top of the rebuttal. To address the concerns on RIP assumption being too strong, we have
performed additional situations when RIP assumption fails. Consider the setting given in
lines 322-332, with rows of $\mathbf{X}$ now sampled from $N(\mathbf{0}, \Sigma)$, with $\Sigma = (1 - \mu)\mathbf{I} + \mu \mathbf{1}\mathbf{1}^\mathsf{T}/d$.
On the right, we plot simulation results with $\mu = 0$ (RIP holds) and $\mu = 0.5$ (RIP fails). We
see that even when RIP fails, our method still exhibits correct rates and outperforms the lasso
when the phase transition happens. The gap between gradient descent and the oracle method

is visible due to the $\log k$ factor in Corollary 3, suggesting also that the rate given in Corollary 3 could be tight. We
will address the reviewers concerns by adding a section on potential improvements with an expansion of the above
discussion. We will also compare and contrast RIP and RE assumptions. If space permits, we will also slightly expand
on the intuition.

We have previously attributed the quadratic sample complexity in $k$ to our bounds
being $\ell_\infty$ (which is harder) rather than $\ell_2$. Our focus has been on *minimax-rates and*
*dimension-independent rates* with *optimal computational complexity*. Also, while there
is loss in sample complexity, there is gain in performance that is impossible to be
achieved by the lasso (see Corollary 3 and lines 334-339).

We stick to the simulation setting described in lines 308-313 and 322-328, with $d = $
5000. The left figure on the right compares $\ell_2$ error ratios for gradient descent and
lasso. The blue region corresponds to our method achieving lower error, while the red
region corresponds to the lasso achieving lower error than gradient descent. This plot strongly suggests, that sample
complexity linear in $k$ should indeed be enough to match/exceed performance of the lasso. The question remains,
whether the $\ell_\infty$ bounds in Theorems 1 and 3 (in particular for stopping time $t$, $\|\mathbf{w}_t \odot \mathbf{1}_{S^c}\|_\infty \leq \sqrt{\alpha}$) require sample
complexity quadratic in $k$? The figure on the right side suggests that the sample complexity linear in $k$ is enough to
satisfy even the $\ell_\infty$ bounds. We expect this sample complexity gap to be addressed in future work.

Given the results above, we absolutely agree with the suggestion to include a discussion on sub-optimal sample
complexity in our revision. We plan to do so in an extra section on potential improvements (see also response to R#2).

[Meta-Review · NeurIPS 2019]

The authors study a reparametrization of the least squares problem such that early-stopped gradient descent mimicks L1 penalization. The idea is very creative and the reviewers were mostly positive. However, several important critiques were raised. I myself am also leaning towards the positive, but want to see the paper improved in several places in a camera-ready version. The intuition for the reparametrization needs to be much more prominent. Also, the main pros and cons of the new proposal could be more properly highlighted. Is this just a method that is restricted in a sense to be used under RIP, practically speaking? Or should we think of this as a general-purpose tool (like the lasso), with theory that describes it behavior in idealized cases? Also, the presentation in the experimental section can be improved; the figures are currently way too small to read properly. Also, the coordinate descent (aka forward stagewise) view should be more properly highlighted, explained, and comapred. This is of course the main "competitor" in that it is a simple iterative algorithm that stopped early produces something like L1 regularization. In addition to the references given in the related work section, the authors should pay attention to Tibshirani (2015), "A General Framework for Fast Stagewise Algorithms" where the connection between stagewise and L1 regularization is clearly/intuitively explained.